# ME31B globally represses maternal mRNAs by two distinct mechanisms during the *Drosophila* maternal-to-zygotic transition

Miranda Wang[1,2†], Michael Ly[1†], Andrew Lugowski[1,2], John D Laver[2], Howard D Lipshitz[2], Craig A Smibert[2,3], Olivia S Rissland[1,2,4,5‡*]

[1]Molecular Medicine Program, The Hospital for Sick Children Research Institute, Toronto, Canada; [2]Department of Molecular Genetics, University of Toronto, Toronto, Canada; [3]Department of Biochemistry, University of Toronto, Toronto, Canada; [4]Department of Biochemistry and Molecular Genetics, University of Colorado Denver School of Medicine, Aurora, United States; [5]RNA Bioscience Initiative, University of Colorado Denver School of Medicine, Aurora, United States

**Abstract** In animal embryos, control of development is passed from exclusively maternal gene products to those encoded by the embryonic genome in a process referred to as the maternal-to-zygotic transition (MZT). We show that the RNA-binding protein, ME31B, binds to and represses the expression of thousands of maternal mRNAs during the *Drosophila* MZT. However, ME31B carries out repression in different ways during different phases of the MZT. Early, it represses translation while, later, its binding leads to mRNA destruction, most likely as a consequence of translational repression in the context of robust mRNA decay. In a process dependent on the PNG kinase, levels of ME31B and its partners, Cup and Trailer Hitch (TRAL), decrease by over 10-fold during the MZT, leading to a change in the composition of mRNA–protein complexes. We propose that ME31B is a global repressor whose regulatory impact changes based on its biological context.
DOI: https://doi.org/10.7554/eLife.27891.001

*For correspondence: olivia.rissland@gmail.com

†These authors contributed equally to this work

Present address: ‡Department of Biochemistry and Molecular Genetics, University of Colorado Denver School of Medicine, Aurora, United States

**Competing interests:** The authors declare that no competing interests exist.

## Introduction

A connection between translational regulation and mRNA degradation is a central principle of post-transcriptional regulation. Regulation of both processes relies upon the 5′ cap and 3′ poly(A) tail together with associated proteins: eIF4E (which binds the cap), PABP (which binds the 3′ tail), and the translation initiation factor eIF4G (*Gallie, 1991*; *Decker and Parker, 1993*; *Caponigro and Parker, 1995*; *Coller et al., 1998*; *Schwartz and Parker, 2000*; *Kahvejian et al., 2005*). Their connection, however, extends far beyond merely using the same factors. Across different regulatory pathways and biological contexts, a recurring theme is that translational repression precedes and likely causes mRNA decay (*Radhakrishnan and Green, 2016*). For instance, some regulatory factors, such as microRNAs (miRNAs), repress translation of their targets before initiating decay (*Bazzini et al., 2012*; *Béthune et al., 2012*; *Djuranovic et al., 2012*), and, in other cases, broadly blocking translation initiation stimulates mRNA decay (*Schwartz and Parker, 1999*). Nonetheless, much about the link between translational repression and mRNA decay remains unknown.

A conserved DEAD-box helicase has emerged as a central player in both processes (*Nakamura et al., 2004*; *Coller and Parker, 2005*; *Chu and Rana, 2006*). Known as ME31B (in *Drosophila*), Dhh1 (in yeast), and DDX6 (in vertebrates), this helicase both represses translation and stimulates mRNA decay. For instance, in the *Drosophila* embryo, ME31B represses translation via a

factor called Cup, which interacts with ME31B through an adaptor protein called Trailer Hitch (TRAL) (*Nakamura et al., 2004*; *Tritschler et al., 2008*). Because Cup directly binds eIF4E at the same site as eIF4G, the Cup–TRAL–ME31B complex disrupts the eIF4E–eIF4G interaction and inhibits translation initiation (*Wilhelm et al., 2003*; *Nakamura et al., 2004*; *Nelson et al., 2004*; *Kinkelin et al., 2012*). Although less understood mechanistically, orthologs also repress translation in yeast, *Plasmodium*, *Xenopus laevis*, *Caenorhabditis elegans*, and humans (*Minshall et al., 2001*; *Chu and Rana, 2006*; *Mair et al., 2006*; *Boag et al., 2008*; *Sweet et al., 2012*).

ME31B/DDX6/Dhh1 has been linked to several different mRNA decay pathways. In yeast, Dhh1p is critical for decay of mRNAs with suboptimal codons (*Radhakrishnan et al., 2016*). In metazoa, ME31B/DDX6 is important for miRNA-mediated decay where ME31B is thought to be recruited through its interaction with CNOT1, a major component of the CCR4-NOT deadenylase component (*Chu and Rana, 2006*; *Chen et al., 2014b*; *Mathys et al., 2014*; *Rouya et al., 2014*). Once bound, ME31B, as well as its orthologs, then links the decapping and deadenylation machinery. In *Drosophila*, ME31B does so through a set of conserved interactions with two additional adaptor proteins, Enhancer of decapping 3 (EDC3) and HPat (also known as Patr-1 or Pat1), which in turn interact with the decapping complex (*Kshirsagar and Parker, 2004*; *Haas et al., 2010*; *Jonas and Izaurralde, 2013*). In addition to directly interacting with Cup, TRAL is also able to bind DCP1 (*Wilhelm et al., 2005*; *Tritschler et al., 2008*, *2009*), and so, when Cup expression or binding is reduced, the binding of TRAL to ME31B may also trigger mRNA decay.

The early *Drosophila* embryo and *Xenopus* oocyte have provided powerful biochemical and genetic contexts to dissect the molecular mechanisms of translational repression for two reasons (*Castagnetti et al., 2000*; *Minshall and Standart, 2004*; *Nelson et al., 2004*; *Jeske et al., 2011*). First, they have robust translational control but lack mRNA decapping (*Gillian-Daniel et al., 1998*; *Zhang et al., 1999*; *Minshall et al., 2001*; *Subtelny et al., 2014*; *Eichhorn et al., 2016*). Second, both contexts have little or no transcription, and so post-transcriptional regulation of mRNA translation, stability, and sub-cellular localization plays a particularly important role in coordinating gene expression (*Varnum and Wormington, 1990*; *Braat et al., 2004*; *Nakamura et al., 2004*; *Kim and Richter, 2006*; *Laver et al., 2015b2015b*).

Many critical developmental events, such as the maternal-to-zygotic transition (MZT), occur during early *Drosophila* embryogenesis. The MZT marks the clearance of maternally deposited mRNAs and the initiation of zygotic transcription (*Tadros and Lipshitz, 2009*). In *Drosophila*, RNA binding proteins, such as Smaug (SMG) and Brain Tumor (BRAT), participate in clearance of maternal mRNAs (*Gerber et al., 2006*; *Tadros et al., 2007*; *Chen et al., 2014a*; *Laver et al., 2015a*). The tightly controlled expression of each of these proteins depends upon developmental events such as egg activation—which stimulates the Pan gu (PNG) kinase—and zygotic transcription (*Tadros et al., 2007*; *Vardy and Orr-Weaver, 2007*; *Benoit et al., 2009*; *Luo et al., 2016*).

ME31B, TRAL, and Cup were originally identified because of their essential developmental functions in *Drosophila* oogenesis and embryogenesis (*Keyes and Spradling, 1997*; *Nakamura et al., 2001*; *Wilhelm et al., 2003*; *Nakamura et al., 2004*; *Nelson et al., 2004*; *Wilhelm et al., 2005*). One of their key targets is *oskar* (*osk*) mRNA, which is transported from nurse cells to the posterior pole of the oocyte in a translationally repressed state (*Nakamura et al., 2001*; *Wilhelm et al., 2003*). However, the extent to which this complex regulates translation more broadly in the oocyte and early embryo remains unknown.

Here, we sought to understand translational repression in the early *Drosophila* embryo on a transcriptome-wide scale. We found that Cup, TRAL, and ME31B bind nearly all expressed transcripts in a complex that also contains eIF4E and PABP. Prior to zygotic genome activation, ME31B binding is associated with translational repression. During the MZT, the abundance of Cup, TRAL, and ME31B falls, and binding of the remaining ME31B to mRNAs instead triggers their destruction. We propose that ME31B is a general regulatory factor whose repressive function can manifest differently depending on its biological context.

# Results

## PABP interacts with ME31B, TRAL, and Cup in the early embryo

Because PABP and eIF4G are deeply connected with both mRNA translation and stability, we first analyzed the proteins present in PABP or eIF4G immunoprecipitates by tandem mass spectrometry. We compared the composition of these complexes in early embryos (0–1 hr after egg-laying) with those from *Drosophila* S2 cells, a highly differentiated, macrophage-like cultured cell line (*Figure 1A,B*; *Figure 1—source data 1*). As expected (*Imataka et al., 1998*), PABP and eIF4G were abundant and co-associated in S2 cells where they interacted with a broadly similar set of proteins, including eIF4E and other translation initiation factors.

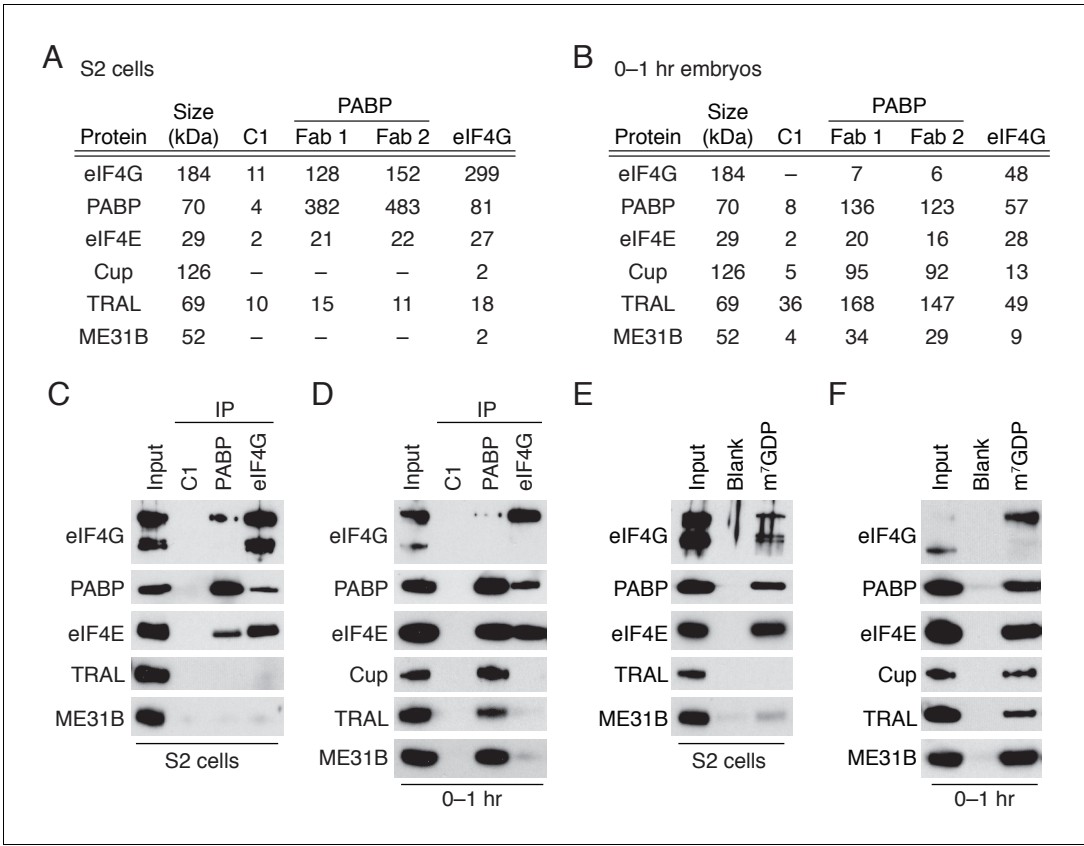

**Figure 1.** A complex containing eIF4E, Cup, TRAL, ME31B, and PABP is abundant in 0–1 hr embryos. (**A**) Selected results from mass spectrometry of PABP and eIF4G co-immunoprecipitations in S2 cells. Extracts from S2 cells were immunoprecipitated with Fab C1 (a negative control), anti-PABP or anti-eIF4G Fabs. Shown are the peptide counts and protein size for proteins of interest. (**B**) As in (**A**), but for extracts from 0 to 1 hr embryos. (**C**) Western blot analysis for proteins co-immunoprecipitated with PABP and eIF4G in S2 cells. Western blots of input and immunoprecipitates were probed for the indicated proteins. (**D**) As in (**C**), but for extracts from 0 to 1 hr embryos. (**E**) Proteins binding m$^7$GDP beads in S2 cell extracts. Lysates were incubated with either blank agarose beads or beads conjugated with m$^7$GDP. Western blots of the input and bound fractions are shown for the indicated proteins. (**F**) As in (**E**), except with 0–1 hr embryo extracts.
DOI: https://doi.org/10.7554/eLife.27891.002

The following source data and figure supplement are available for figure 1:

**Source data 1.** Mass spectrometry results for PABP and eIF4G immunoprecipitations in S2 cells.
DOI: https://doi.org/10.7554/eLife.27891.004

**Source data 2.** Mass spectrometry results for PABP and eIF4G immunoprecipitations in 0–1 hr embryos.
DOI: https://doi.org/10.7554/eLife.27891.005

**Figure supplement 1.** ME31B interacts with PABP and eIF4E in the early embryo.
DOI: https://doi.org/10.7554/eLife.27891.003

In contrast, distinct sets of proteins were associated with PABP and eIF4G in the 0–1 hr embryos (*Figure 1B*; *Figure 1—source data 2*). While eIF4G-containing complexes resembled those found in S2 cells, PABP co-immunoprecipitated ME31B, Cup, and TRAL, proteins that were not present in the PABP or eIF4G complexes from S2 cells. Interestingly, proteins proposed to stabilize ME31B and Cup on mRNAs were notably absent (*e.g.*, BRU1 [*Nakamura et al., 2004*]) or were present only at low levels (e.g., SMG [*Jeske et al., 2011*]) in our mass spectrometry results (*Figure 1—source data 2*). Less eIF4G co-immunoprecipitated with PABP in 0–1 hr embryos compared to S2 cells, although PABP was abundant in eIF4G complexes in both cell types. These data suggest that, in the early embryo, a substantial fraction of PABP complexes do not contain eIF4G but, instead, contain ME31B, TRAL, and Cup. In contrast, most eIF4G complexes contain PABP.

In order to further assess these differences between early embryos and S2 cells, we analyzed the immunoprecipitated complexes by immunoblotting. As expected, PABP and eIF4G co-immunoprecipitated with each other and eIF4E in S2 cells, but neither immunoprecipitate contained detectable ME31B and TRAL (*Figure 1C*; Cup is not abundant in S2 cells). In contrast, immunoblot analysis of 0–1 hr embryos detected stable association of PABP with eIF4E, Cup, TRAL, and ME31B. Because the interaction between ME31B and PABP in the early embryo was sensitive to RNase (*Figure 1—figure supplement 1*), it is likely to be indirect. Despite the fact that little eIF4G was detected in the PABP immunoprecipitates from early embryos (*Figure 1B*), as in S2 cells, embryonic eIF4G complexes contained eIF4E and PABP (*Figure 1D*).

Reciprocal co-immunoprecipitation/western blot experiments gave similar results. In S2 cells, as expected, ME31B failed to interact with PABP, eIF4E, and eIF4G, although the co-immunoprecipitates did contain known interacting proteins, such as EDC3 and HPat (*Figure 1—figure supplement 1*). To probe embryonic ME31B interactions in 0–1 hr embryos, we used a previously developed *eGFP-ME31B* trap line (*Buszczak et al., 2007*) and found that, consistent with the PABP co-immunoprecipitations, ME31B interacted with Cup, TRAL, PABP, and eIF4E, although not eIF4G (*Figure 1—figure supplement 1*).

eIF4E specifically recognizes the 7-methylguanosine cap of mRNAs. We took advantage of the specific binding of eIF4E to the cap analog 7-methyl-GDP ($m^7$GDP) to test whether the eIF4E, Cup, TRAL, and ME31B can assemble through protein-protein interactions. To assess non-specific binding, we also analyzed the proteins bound to agarose beads. When the cap analog was incubated with S2 lysates, eIF4E, eIF4G, and PABP all bound selectively, but neither TRAL nor ME31B was enriched in the bound fraction (*Figure 1E*), presumably because of the relatively low expression of Cup in these cells. In contrast, when this experiment was carried out in 0–1 hr embryo lysates, Cup, TRAL, and ME31B bound $m^7$GDP in addition to eIF4E, PABP, and eIF4G (*Figure 1F*), consistent with the hypothesis that protein-protein interactions are sufficient to drive assembly of the eIF4E–Cup–TRAL–ME31B complex in the early embryo.

Together, our data indicate that the *Drosophila* early embryo possesses two distinct complexes containing eIF4E and PABP: one that contains eIF4G, and another that contains ME31B, TRAL, and Cup but not eIF4G. These distinct complexes support the proposal that eIF4E cannot bind both eIF4G and Cup simultaneously (*Kinkelin et al., 2012*). Our data also demonstrate that Cup, TRAL, and ME31B suffice to form a stable complex with cap-bound eIF4E. In vivo, additional mRNA sequences and proteins such as BRU1 or SMG (*Wilhelm et al., 2003*; *Nakamura et al., 2004*; *Nelson et al., 2004*; *Piccioni et al., 2009*) may stabilize or enhance formation of this complex on particular mRNAs.

## PABP and ME31B bind similar transcripts in the early embryo

Do ME31B and PABP bind the same mRNAs? To identify mRNAs bound to each protein, we immunoprecipitated ME31B or PABP from 0 1 hr embryo lysates and measured RNA abundance by RT-qPCR or high-throughput sequencing ('RIP-seq'; see below). To minimize the possibility of dissociation and reassortment of protein–RNA interactions, our protocol contained shortened incubation times (compared to standard immunoprecipitation protocols). A benchmark transcript, *Act5C*, showed significant enrichment in both our PABP and ME31B immunoprecipitations compared to control pull-downs (*Figure 2A*; 1,189- and 42,000-fold, respectively), indicating that less than 0.1% of *Act5C* in either immunoprecipitation was attributable to background binding.

We analyzed RNA abundance in the total and immunoprecipitated samples transcriptome-wide with RNA sequencing (*Figure 2—figure supplement 1*). Because the amount of each transcript in

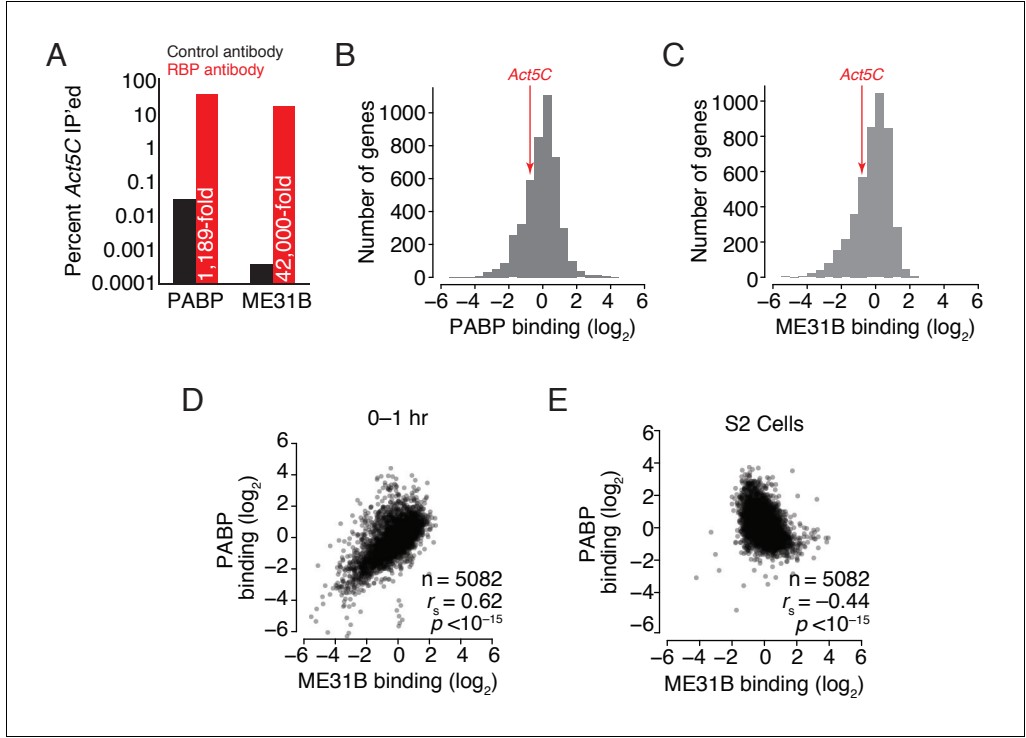

**Figure 2.** ME31B and PABP bind similar mRNAs in 0–1 hr embryos. (**A**) Enrichment of *Act5C* transcripts in PABP and ME31B immunoprecipitations. Extracts from 0 to 1 hr embryos were immunoprecipitated with the indicated antibodies (control pull-downs, black; PABP or ME31B pull-downs, red). In the case of PABP, complexes were immunoprecipitated with Fab1; ME31B was precipitated from *eGFP-ME31B* embryos with anti-GFP antibodies. The percent of *Act5C* mRNA pulled down was quantified by RT-qPCR, primed by oligo(dT) (in the case of PABP) or random-hexamers (in the case of ME31B). Numbers reflect the fold enrichment of *Act5C* in PABP or ME31B immunoprecipitations relative to the control immunoprecipitations. (**B**) Distribution of PABP enrichment values transcriptome-wide, as determined by RIP-seq. For mRNAs of each gene that satisfied the expression cut-offs, PABP binding was calculated. Highlighted is the binding for *Act5C*. (**C**) As in (**B**), except for ME31B binding values. (**D**) Comparison of PABP and ME31B binding in 0–1 hr embryos. (**E**) Comparison of PABP and ME31B binding in S2 cells.

DOI: https://doi.org/10.7554/eLife.27891.006

The following figure supplements are available for figure 2:

**Figure supplement 1.** ME31B and PABP bind similar transcripts in the early embryo.
DOI: https://doi.org/10.7554/eLife.27891.007
**Figure supplement 2.** ME31B and PABP binding as measured by RT-qPCR and RNA-sequencing.
DOI: https://doi.org/10.7554/eLife.27891.008

the bound samples is influenced both by its interaction with ME31B or PABP and by its abundance, we normalized the FPKM values in the immunoprecipitated sample with that from the input sample to control for expression differences between different genes. This normalization procedure provides relative, not absolute, binding, but we will refer to these as 'binding' values for simplicity. We compared the RIP-seq binding measurements with the fraction of the total mRNA immunoprecipitated (as measured by RT-qPCR) and found agreement between these two measurements across a range of PABP and ME31B binding values for several transcripts (*Act5C*, *veli*, *ana2* and *gnu*; see *Figure 2—figure supplement 2*).

Based on the enrichment of the benchmark *Act5C* transcript (as determined by RT-qPCR) and its binding (as determined by sequencing), the vast majority of expressed genes were enriched in the PABP pull-downs (*Figure 2B*). Similarly, despite the fact that only a few targets of ME31B have been described in oogenesis and embryogenesis (e.g., *grk*, *osk*, *nos*) (*Nakamura et al., 2001*; *Wilhelm et al., 2005*; *Jeske et al., 2011*), transcripts from nearly all expressed genes were enriched in the ME31B pull-down (*Figure 2C*). For instance, at 0–1 hr, the $\log_2$ binding of *nos* transcripts was

–0.323 while the median $\log_2$ binding of all transcripts was 0, indicating that mRNAs from more than half of the expressed genes were bound more highly by ME31B than this well-characterized target.

In order to address whether transcripts with very low binding values were bound by ME31B and PABP, we used RT-qPCR to determine the enrichment of *smg* mRNA in both immunoprecipitations relative to IgG. In our RIP-seq data, ME31B and PABP binding to *smg* was in the bottom 1% to 2% of all expressed transcripts (with a $\log_2$ value of –3.5 and –2.6, respectively). Nonetheless, in immunoprecipitations of both proteins, *smg* mRNA was enriched by at least 10-fold in three biological replicates, as measured by RT-qPCR (*Figure 2—figure supplement 2*), indicating that even the most poorly bound mRNAs are still enriched in ME31B and PABP immunoprecipitations. Thus, although PABP and ME31B are capable of binding transcripts from most expressed genes, they do so to vastly differing extents.

We next compared the relationship between ME31B and PABP binding. The uncorrected RNA abundances in the ME31B and PABP immunoprecipitations were significantly correlated in both 0–1 hr embryos and S2 cells, although to a lesser extent for the latter (*Figure 2—figure supplement 1*; $r_s = 0.94$ and $r_s = 0.63$, respectively). Consistent with a detectable interaction between PABP and ME31B in 0–1 hr embryos (*Figure 1*), even after we corrected for total RNA abundance, their binding was still strongly correlated (*Figure 2D*; $r_s = 0.62$, $p<10^{-15}$). In contrast, once we corrected for RNA expression in S2 cells, their binding was now negatively correlated (*Figure 2E*; $r_s = –0.44$, $p<10^{-15}$). This observation is in line with a lack of detectable protein-level interactions between ME31B and PABP in this cellular context and their distinct roles in post-transcriptional regulation (*Rissland, 20162016*). We conclude that the observed interaction between ME31B and PABP in 0–1 hr embryos is explained by co-binding the same transcripts from many different genes rather than being due to both proteins co-binding a small number of specific transcripts, such as *nos* or *osk*.

## ME31B binding is associated with translational repression in the early embryo

Having established that ME31B binds to many transcripts in the early embryo, we next investigated its impact on gene expression, focusing on its relationships with poly(A)-tail length and translational repression. Consistent with a previous study (*Eichhorn et al., 2016*), we found that in 0–1 hr embryos, mRNAs with long poly(A) tails were more highly translated than those with short tails (*Figure 3A*; $r_s = 0.38$, $p<10^{-15}$). High ME31B binding, however, was associated with short poly(A) tails and low translational efficiency (*Figure 3B,C*; $r_s = –0.38$, $p<10^{-15}$; $r_s = –0.29$, $p<10^{-15}$, respectively).

These correlations might reflect preferential binding of ME31B to mRNAs with short poly(A) tails, whose translational efficiency is inherently low in the early embryo (*Figure 3A,B*). Alternatively, ME31B binding might reduce translational efficiency independent of poly(A)-tail length. To distinguish between these possibilities, we examined the relationship between translational efficiency and ME31B binding after correcting for the length of the poly(A) tail. Although weaker than the uncorrected case, we still observed a significant negative correlation between ME31B and translational efficiency (*Figure 3D*; corrected $r_s = –0.19$, $p<10^{-15}$). Similarly, the negative correlation between poly(A)-tail length and translational efficiency remained significant even after correcting for ME31B binding (*Figure 3E*; corrected $r_s = –0.26$, $p<10^{-15}$).

Many genes displayed a reciprocal relationship between ME31B binding and poly(A)-tail length (*Figure 3A–C*, teal dots). For instance, *CycA*, *CycB*, and *CycB3* had long poly(A) tails, were poorly bound by ME31B, and were highly translated. In contrast, *Debcl* and *Myt1* had short poly(A) tails, were highly bound by ME31B, and were poorly translated. Nonetheless, there were examples of genes where the relationship between tail length and ME31B binding was less straightforward, such that their translation appeared to be predominantly regulated by one or the other. One such example was *nos* (*Figure 3A–C*, pink dot). Despite being a known ME31B target in the early embryo (*Jeske et al., 2011*), its binding at 0–1 hr, as noted above, was near the median of all genes. On the other hand, *nos* was one of the most poorly translated transcripts with over 30-fold lower translation relative to other genes with comparable ME31B binding. Its low translation efficiency was instead explained by a very short poly(A) tail, with a length of 26 nt on average. There were also examples of genes, such as *Rrp45*, whose translation was most explained by ME31B binding rather than tail length (*Figure 3A–C*, blue dot). Together these data suggest that ME31B binding and poly(A)-tail

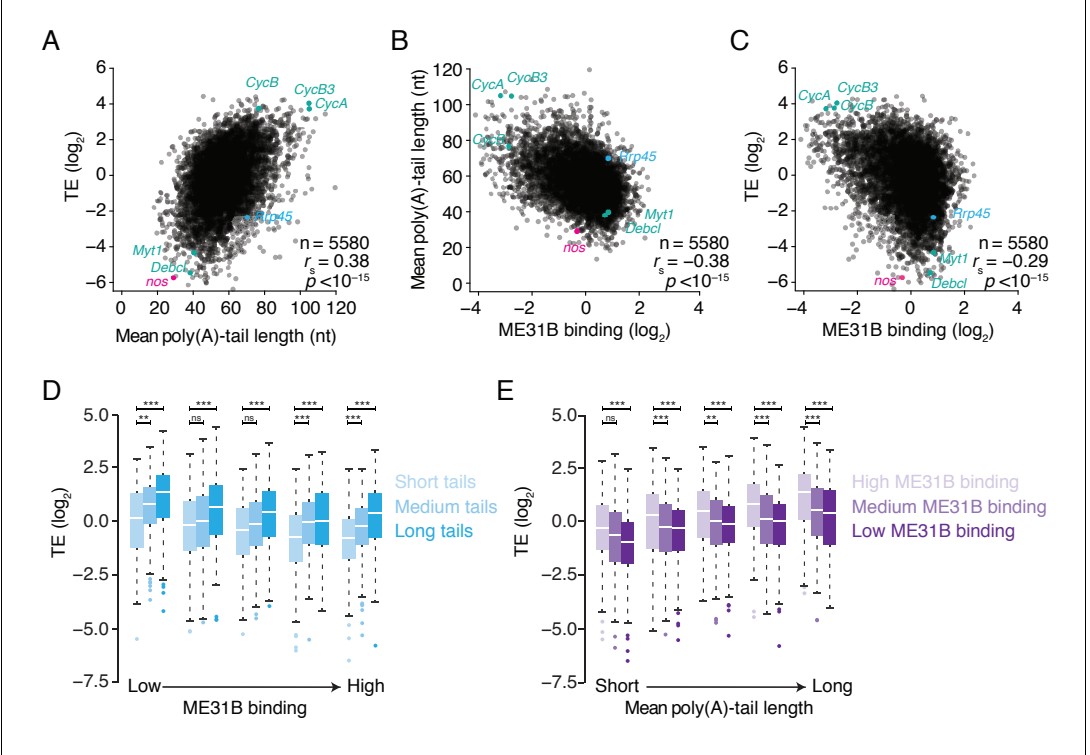

**Figure 3.** ME31B binding is associated with repressed translation in 0–1 hr embryos. (A) The relationship of poly(A)-tail length and translational efficiency (TE) in 0–1 hr embryos. Plotted are the mean poly(A)-tail lengths and TE values for each gene expressed above cut-off values in 0–1 hr embryos. Select genes concordantly regulated by poly(A)-tail length and ME31B are shown in teal; *nos*, in pink; *Rrp45*, in blue. (B) The relationship of poly(A)-tail length and ME31B binding, otherwise as in (A). (C) The relationship of ME31B binding and translational efficiency (TE), otherwise as in (A). (D) The effects of ME31B and poly(A)-tail length on translation. Transcripts were binned into quintiles by their ME31B binding and then into thirds by their mean poly(A)-tail length. Box and whisker plots of translational efficiencies are plotted (line, median; box, quartiles; whiskers, range, excluding outliers). Significance was calculated by the two-tailed Kolmogorov-Smirnov test. **$p<0.01$; ***$p<0.001$; ns, not significant. (E) The effects of ME31B and poly(A)-tail length on translation. Transcripts were binned into quintiles by their mean poly(A)-tail length and then into thirds by their ME31B binding, otherwise as in E.

DOI: https://doi.org/10.7554/eLife.27891.009

The following figure supplement is available for figure 3:

**Figure supplement 1.** Relationship of PABP binding and poly(A) tail length.

DOI: https://doi.org/10.7554/eLife.27891.010

length, although often reciprocally interdependent, can also regulate translation independently in the early *Drosophila* embryo.

Given the strong positive relationship between ME31B binding and PABP binding and the negative relationship between ME31B binding and poly(A)-tail length, we were surprised to find no significant relationship between PABP binding and poly(A)-tail length ($r_s = -0.01$, p=0.42; ***Figure 3— figure supplement 1***). We hypothesized that this result might be explained by the existence of PABP in two distinct complexes: one with ME31B and the other without, each with distinct poly(A)-tail length profiles. In order to infer the relationship of PABP binding and poly(A)-tail length for the complexes lacking ME31B, we examined the enrichment of PABP binding relative to ME31B binding (***Figure 3—figure supplement 1***). In 0–1 hr embryos, this enrichment was strongly correlated with poly(A)-tail length, even when we corrected for the underlying negative relationship between ME31B binding and tail length ($r_s = 0.47$, p<$10^{-15}$; corrected $r_s = 0.37$, p<$10^{-15}$). These data suggest that longer poly(A) tails recruit more PABP, but only when these transcripts are not bound by ME31B.

Taken together, our transcriptome-wide analyses indicate that poly(A)-tail length and ME31B binding shape translation in the early embryo, likely by different mechanisms. Many genes (e.g.,

*CycB* and *Myt1*) appear use both, and so the observed correlations reflect lower, rather than upper, bounds for the contributions of each pathway.

## ME31B, TRAL, and Cup are regulated by multiple concordant mechanisms during the maternal-to-zygotic transition

We next measured the abundance of eIF4E, eIF4G, PABP, Cup, ME31B, TRAL, EDC3, and HPat across the first five hours of development (*Figure 4A*). The levels of eIF4E, eIF4G, PABP, and HPat did not change and were similar to those seen in S2 cells, while eIF4E decreased only slightly over the time-course. In contrast, ME31B, TRAL, Cup, and EDC3 protein abundance decreased sharply; the largest decrease occurred between the 1–2 hr and 2–3 hr time points, coinciding with the

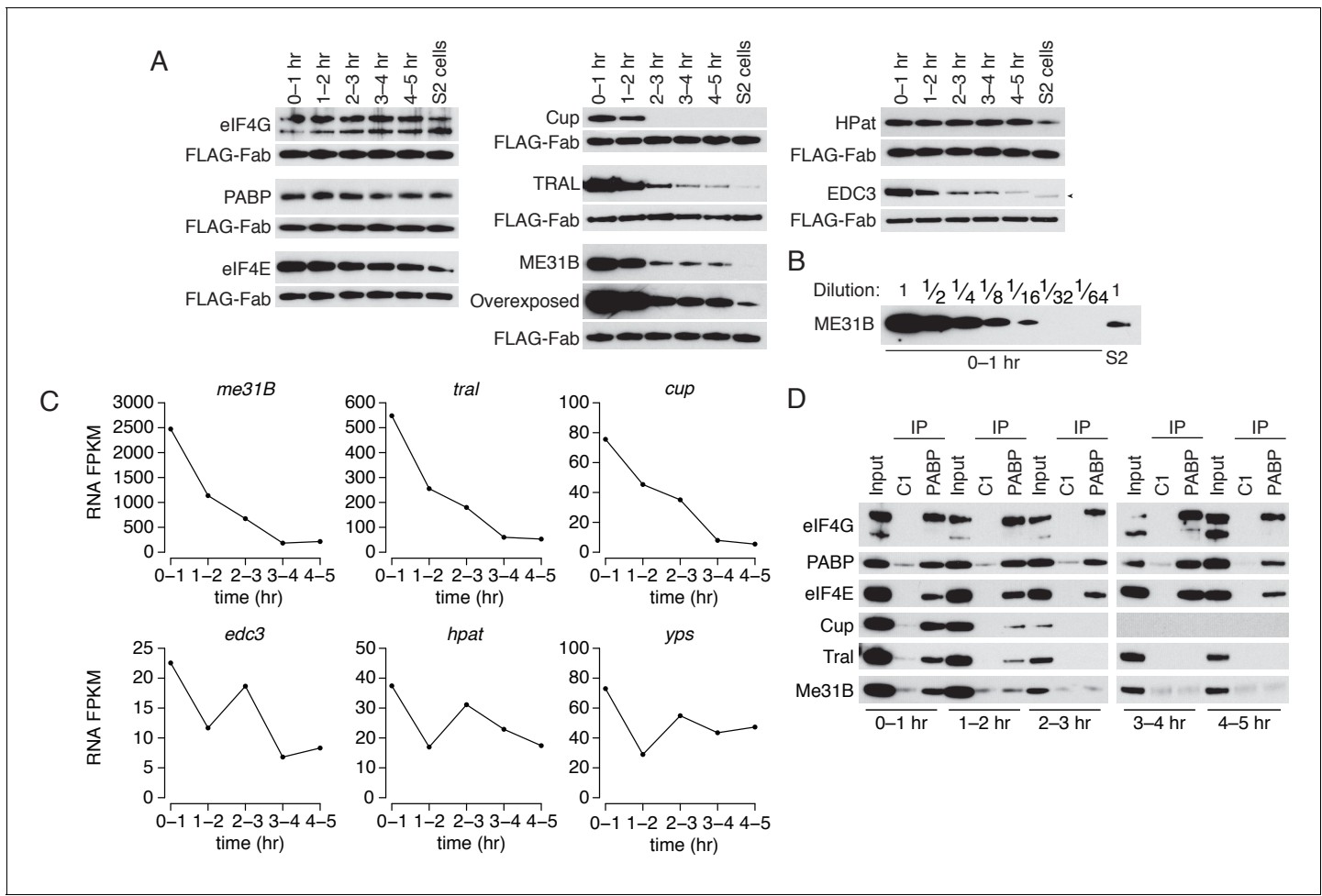

**Figure 4.** The levels of ME31B, TRAL, and Cup are developmentally regulated through multiple mechanisms. (**A**) Western blot analysis of protein abundance over the first five hours of development. Extracts from embryos harvested at one-hour intervals and from S2 cells were collected, and constant amounts of exogenous FLAG-tagged Fabs were added to each lysate. Western blot analysis of the indicated proteins is shown. The arrowhead indicates a nonspecific band. (**B**) Levels of ME31B in 0–1 hr embryos relative to S2 cells. Western blot of serial dilutions of 0–1 hr embryo extracts and undiluted S2 cell lysate was probed for ME31B. (**C**) Analysis of mRNA levels over the first five hours of development. RNA-seq was used to analyze RNA abundance in embryos harvested at one-hour intervals. Libraries were prepared with rRNA-depletion. Shown are the FPKM values for the indicated genes. (**D**) Analysis of the interaction between PABP and Cup, TRAL, and ME31B in early development. Extracts from embryos at the indicated time points were immunoprecipitated with control C1 or anti-PABP Fabs. Western blots of inputs and immunoprecipitates were probed using the indicated antibodies.

DOI: https://doi.org/10.7554/eLife.27891.011

The following figure supplement is available for figure 4:

**Figure supplement 1.** Regulation of mRNA and protein abundances during early development.
DOI: https://doi.org/10.7554/eLife.27891.012

initiation of high zygotic transcription. Transgenic *eGFP-ME31B* embryos showed similar kinetics (*Figure 4—figure supplement 1*). We also noted that Cup, ME31B, TRAL, and EDC3 were significantly more abundant in the 0–1 hr embryos than in S2 cells, and quantitative western blotting showed that ME31B proteins levels were 15–20 fold higher in these embryos than in S2 cells (*Figure 4B*).

Previous proteomic studies have also found that Cup, TRAL, and ME31B decrease at the MZT (*Sysoev et al., 2016*). Our re-analysis of those data showed that the decreases in Cup, ME31B, and TRAL levels rank among the largest reductions in abundance (*Figure 4—figure supplement 1*). In fact, of the more than 1000 proteins that they measured, the only annotated RNA-binding protein that showed a greater reduction is the cytoplasmic poly(A) polymerase Wispy (*Figure 4—figure supplement 1*), which is known to be dynamically controlled during embryogenesis (*Cui et al., 2008*).

We next examined the abundance of the corresponding mRNAs. Like their protein abundances, *me31b*, *tral*, and *cup* transcript abundances also decreased during the first five hours of *Drosophila* development such that, by 3–4 hr, less than 10% of the maternally deposited mRNAs remained (*Figure 4C*). Although *eIF4E* and *eIF4G* mRNA abundance also decreased as embryos transitioned from maternal to zygotic mRNA supplies (*Figure 4—figure supplement 1*), eIF4E and eIF4G protein levels remained essentially constant (*Figure 4A*). Moreover, the mRNAs encoding several other RBPs—including *pAbp*, *Edc3*, *HPat*, *yps*, and *twin*—underwent little or no change during the MZT (*Figure 4* and *Figure 4—figure supplement 1*). We conclude that mRNA-binding proteins are not globally cleared during the MZT, but, rather, that both the mRNAs and protein products of *me31b*, *cup*, and *tral* are specifically targeted for destruction during the MZT.

We next investigated the developmental timing of the dissociation of the complexes that contain PABP, ME31B, Cup and TRAL. As before, we detected a robust interaction between PABP and ME31B in 0–1 hr lysates (*Figure 4D*). However, by 1–2 hr, PABP immunoprecipitated less Cup, TRAL, and ME31B, even though there was little change in their protein levels between 0–1 and 1–2 hr. From 2–3 hr onwards, PABP failed to detectably co-immunoprecipitate these three proteins, despite the continuing presence of ME31B and TRAL (albeit at substantially lower levels). In contrast, PABP co-immunoprecipitated eIF4E and eIF4G throughout the entire time-course.

We conclude that the interactions between PABP and the Cup–TRAL–ME31B complex are regulated concordantly during the MZT. Protein and mRNA degradation likely act as major drivers regulating this complex late in the MZT, but its disassembly appears to be a regulatory event that occurs earlier than, and possibly independent of, the substantial reductions at the later time-points.

## Clearance of ME31B, TRAL and Cup depends on the PNG kinase

Because *me31b*, *cup* and *tral* mRNAs all decrease during the MZT, we asked whether their destruction required SMG, an RNA-binding protein that clears hundreds of maternally deposited transcripts from the embryo and that has been reported to bind *me31B* and *tral* transcripts (*Tadros et al., 2007*; *Chen et al., 2014a*). To test this idea, we measured the effect of loss of SMG on mRNA abundance in embryos. Because *smg* mutants show developmental defects after nuclear cycle 11 and fail to cellularize (*Benoit et al., 2009*; *Dahanukar et al., 1999*), we limited our analysis to the first three hours of embryogenesis. The *me31B*, *cup* and *tral* mRNAs were more abundant in *smg* mutants than in wild-type embryos (*Figure 5—figure supplement 1*). However, loss of SMG did not impair destruction of ME31B protein during the MZT (*Figure 5—figure supplement 1*), consistent with the view that distinct mechanisms underlie the destruction of *me31b* mRNA and clearance of ME31B protein.

The PNG kinase has a central role in the oocyte-to-embryo transition, shaping the transcriptome and proteome after egg activation (*Tadros et al., 2003*; *2007*; *Vardy and Orr-Weaver, 2007*; *Kronja et al., 2014a*; *2014b*). We hypothesized that PNG might also be important for the clearance of ME31B, TRAL, and Cup, and so we measured the abundance of ME31B, TRAL, and Cup proteins in *png^{50}* mutant embryos across early development. (*png^{50}* is a weak hypomorphic allele with only mild developmental defects during the first five hours of development). In the absence of PNG, levels of ME31B, TRAL, and Cup failed to decline during the MZT, and clearance of EDC3 was delayed (*Figure 5A*). eIF4E, eIF4G, PABP levels were constant over the time course in both wild-type and *png*-mutant embryos (*Figure 4A* cf. 5A) whereas HPat was cleared in both genotypes.

In order to determine whether the interaction between PABP and the Cup–TRAL–ME31B complex also persisted in *png^{50}* embryos, we performed immunoprecipitations of PABP or eGFP-ME31B

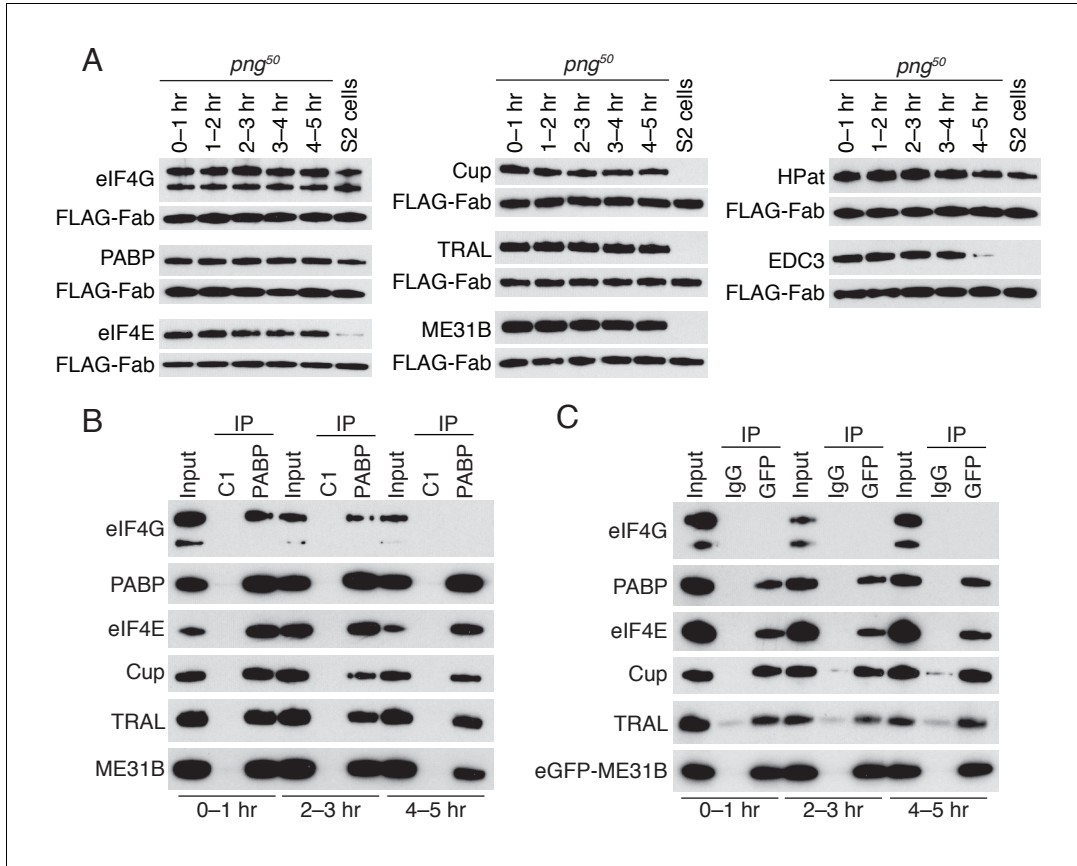

**Figure 5.** PNG is required for the loss of the Cup, TRAL, and ME31B. (**A**) Western blot analysis of *png50* embryos. Extracts were collected at one-hour intervals over the first five hours of development, and a constant amount of exogenous FLAG-tagged Fab was added to each lysate, as in *Figure 4A*. Western blot analyses of the indicated proteins are shown. The arrowhead indicates a nonspecific band. (**B**) Immunoprecipitation of *png50* embryo lysates using control or anti-PABP Fabs. Extracts from 0 to 1, 2–3, and 4–5 hr embryos were co-immunoprecipitated with control C1 or anti-PABP Fabs. Western blots of inputs and immunoprecipitates were probed using the indicated antibodies. (**C**) As in (**B**), except for ME31B. Extracts were isolated from *eGFP-me31B; png50* embryos at 0–1, 2–3, and 4–5 hr after egg laying, and co-immunoprecipitated using rabbit IgG or anti-GFP antibodies. Western blots of inputs and immunoprecipitates were probed using the indicated antibodies.

DOI: https://doi.org/10.7554/eLife.27891.013

The following figure supplement is available for figure 5:

**Figure supplement 1.** Regulation of *me31b*, *tral*, and *cup* transcripts during embryogenesis.

DOI: https://doi.org/10.7554/eLife.27891.014

in the *png50* background and assessed partner proteins by western blotting (*Figure 5B,C*). In contrast to wild-type embryos, where the interaction of PABP with Cup, TRAL, and ME31B was largely lost by 1–2 hr (*Figure 4D*), in *png50* embryos interactions between PABP and Cup, TRAL, and ME31B persisted until 4–5 hr, the last time point we investigated (*Figure 5B,C*). Thus, the PNG kinase is required, either directly or indirectly, for the degradation of ME31B, TRAL, and Cup, as well as for the dissociation of PABP and the Cup–TRAL–ME31B complex during the MZT.

## Transcripts bound by ME31B change during the maternal-to-zygotic transition

To determine how the transcripts bound by ME31B change during the MZT, we performed RIP-seq experiments in lysates from embryos collected at 1 hr intervals over the first five hours of development (*Figure 6*). Consistent with an overall reduction in ME31B protein levels, the amount of our benchmarking transcript, *Act5C*, that co-immunoprecipitated also decreased over the time course

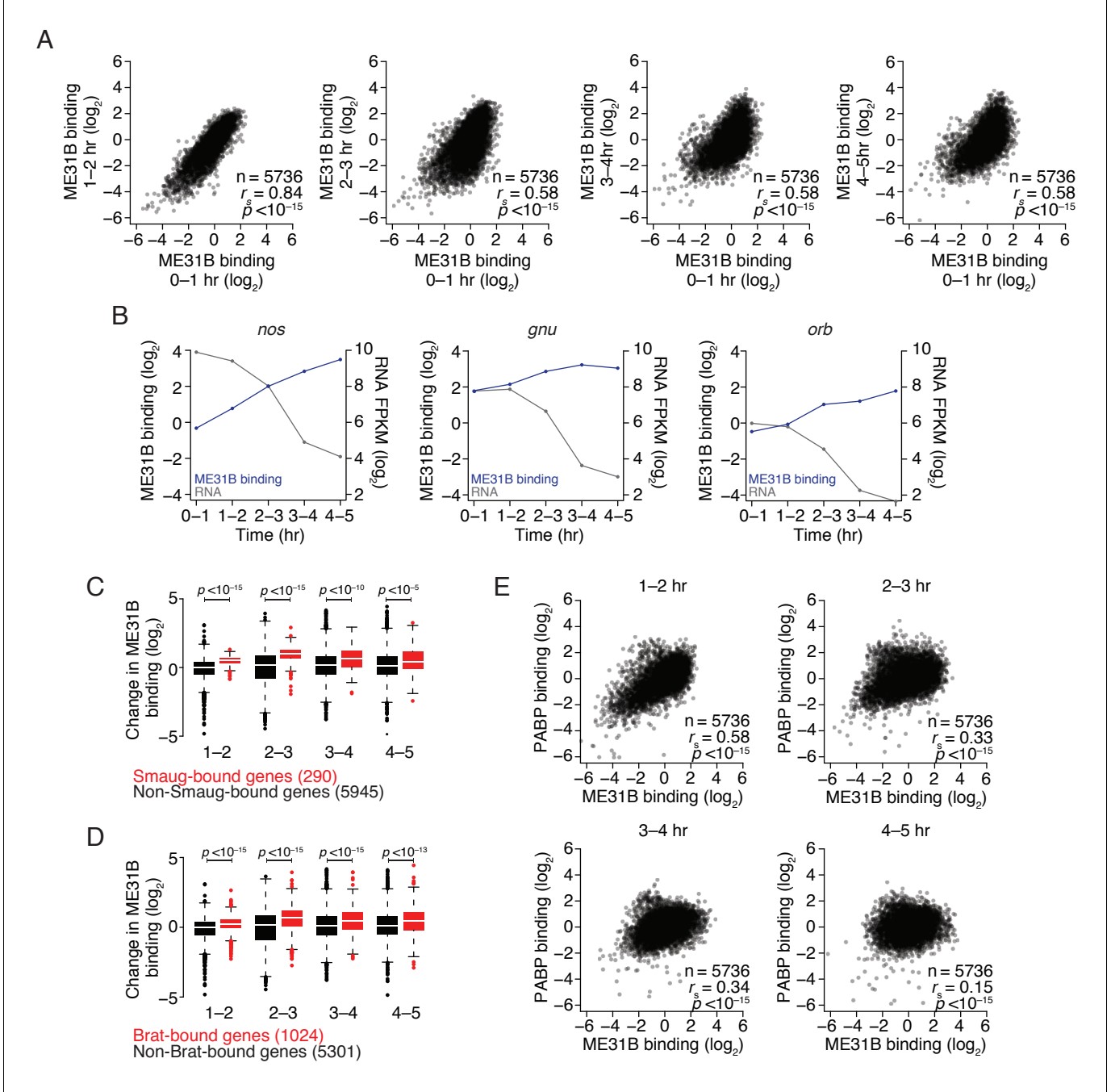

**Figure 6.** The maternal-to-zygotic transition alters the relationship between PABP and ME31B binding. (**A**) Comparison of ME31B binding during early development. Extracts were collected at one-hour intervals for the first five hours of development, and mRNA abundance in the total population as well as those bound by ME31B were determined by sequencing. Note that these libraries were made using rRNA-depletion. Binding for each gene was calculated by normalizing ME31B-bound abundance by its abundance in the total RNA population. Scatter plots for the indicated comparisons are shown. Note that the 0–1 hr sample from this experiment was discussed in *Figure 3*. (**B**) Alterations in ME31B binding and transcript abundance for *nos*, *orb*, and *gnu*. (**C**) A comparison of changes in ME31B binding for SMG-bound mRNAs. For each gene, the fold change in its binding compared to 0–1 hr was calculated. Shown are box-and-whisker plots for SMG-bound genes (in red) and non-SMG bound genes (in black). Line, median; box, quartiles; whiskers, interquartile range. Significance was calculated using the two-tailed Kolmogorov-Smirnov test. (**D**) As in (**C**), except for Brat-bound mRNAs. (**E**) Comparison of PABP and ME31B binding. For each embryonic time point, scatter plots of PABP and ME31B binding are shown.

DOI: https://doi.org/10.7554/eLife.27891.015

The following figure supplements are available for figure 6:

**Figure supplement 1.** Regulation of PABP binding during the MZT.

*Figure 6 continued on next page*

*Figure 6 continued*

DOI: https://doi.org/10.7554/eLife.27891.016

**Figure supplement 2.** Regulation of PABP binding during the MZT.

DOI: https://doi.org/10.7554/eLife.27891.017

(*Figure 6—figure supplement 1*), although thousands of genes were still enriched in the bound fraction at each time point (*Figure 6*). To test this interpretation, we examined four additional transcripts, spanning a range of binding values, by RT-qPCR. All were enriched in ME31B immunoprecipitations relative to IgG immunoprecipitations across the time course, including *smg*, whose binding was in the bottom 10% of associations at 4–5 hr (*Figure 6—figure supplement 1*). We also determined the fraction of these transcripts bound by ME31B at 4–5 hr using RT-qPCR. Although these values were lower than in 0–1 hr embryos by over an order of magnitude, they were in accordance with our RNA-seq measurements (*Figure 6—figure supplement 1cf*. *Figure 2—figure supplement 2*).

In addition to affecting the fraction of transcripts bound by ME31B, the MZT also coincided with a progressive change in the binding of ME31B to different transcripts (*Figure 6A*), such that binding in 0–1 hr embryos, although highly correlated with those in 1–2 embryos ($r_s = 0.83$, $p<10^{-15}$), was more weakly correlated with those at later time points ($r_s = 0.58$, $p<10^{-15}$). As examples, ME31B binding of *ana2* mRNA was high in the early embryo but dropped following the activation of zygotic transcription whereas binding of *smg* mRNA by ME31B increased during early embryogenesis, and peaked at 2–3 hr (*Figure 6—figure supplement 1*).

Because our analysis was restricted to genes defined as expressed at all time points, the observed changes in ME31B binding cannot be explained entirely by the global changes in the transcriptome upon induction of zygotic transcription. The *smg* mRNA, for example, is strictly maternally contributed but binding by ME31B increased during the MZT (*Figure 6—figure supplement 1*). We, therefore, hypothesized that alterations in binding might be explained, at least in part, by differential recruitment of ME31B to specific transcripts by additional *trans*-factors during the MZT. Indeed, RNA-binding proteins, such as SMG, have been proposed to stabilize the Cup–TRAL–ME31B complex on developmentally important transcripts, such as *grk* and *nos* (*Nelson et al., 2004*; *Wilhelm et al., 2005*; *Jeske et al., 2011*). Consistent with a role for SMG in stabilizing ME31B on targets, relative ME31B binding on *nos* increased four-fold from 0 to 1 hr to 2–3 hr (*Figure 6B*), a time period that coincides with the induction of SMG protein (*Smibert et al., 1999*; *Benoit et al., 2009*) and that preceded a sharp decline in *nos* mRNA abundance. When we examined two other SMG targets, *gnu* and *orb*, we observed similar, although less dramatic, trends (*Figure 6B*, *Figure 6—figure supplement 1*).

To determine if this trend held transcriptome-wide, we examined the change in ME31B binding on SMG-bound transcripts (*Chen et al., 2014a*). As with *nos*, relative ME31B binding on SMG-bound mRNAs increased during embryogenesis, and it did so to a greater extent than on non-SMG-bound transcripts (*Figure 6C*; $p<10^{-15}$, two-tailed Kolmogorov-Smirnov [K-S] test). When we repeated this analysis, this time focusing on targets of BRAT (*Laver et al., 2015a*), we found significant recruitment of ME31B to BRAT-bound transcripts (*Figure 6D*; $p<10^{-15}$, K-S test), suggesting that at least two of the major regulatory proteins involved in maternal mRNA clearance recruit and/or stabilize ME31B on their targets, likely in an indirect fashion.

We next performed analogous PABP RIP-seq experiments. Using RT-qPCR, we demonstrated that even transcripts with low PABP binding, such as *smg* mRNA, were enriched by over 10-fold in PABP immunoprecipitations throughout the time course (*Figure 6—figure supplement 2*). As with ME31B, PABP binding globally changed during this time course (*Figure 6—figure supplement 2*), indicating that the MZT coincides with broad changes in the binding of both ME31B and PABP. Interestingly, the relationship we observed in 0–1 hr embryos between PABP binding and poly(A)-tail length also changed during the MZT, such that only a weak correlation between PABP binding and tail length existed by 3–4 hr ($r_s = 0.16$, $p<10^{-15}$).

The relationship between PABP and ME31B binding in 1–2 hr embryos was similar to that seen in 0–1 hr embryos (*Figure 6E cf*. *Figure 2D*; $r_s = 0.62$ vs. $r_s = 0.58$). However, by 2–3 hr, the correlation between PABP and ME31B binding was weaker ($r_s = 0.33$, $p<10^{-15}$) and was even further reduced by 4–5 hr ($r_s = 0.15$, $p<10^{-15}$). Consistent with our inability to detect an interaction between PABP

and ME31B after 2–3 hr (*Figure 4D*), these results indicate that a substantial reorganization of mRNA–protein complex composition occurs during the MZT and that, by the completion of the MZT, PABP and ME31B no longer bind similar sets of transcripts.

## The impact of ME31B on gene expression during the MZT

Finally, we investigated how ME31B binding related to gene expression during the MZT. As described above (*Figure 3C*), ME31B binding negatively correlated with translational efficiency at 0–1 hr ($r_s$ = –0.29). At 2–3 hr, high ME31B binding still correlated with low translational efficiency (*Figure 7A*; $r_s$ = –0.24, p<10$^{-15}$), although the magnitude of its effect was slightly smaller than at 0–1 hr. By 3–4 hr, however, the relationship between ME31B binding and low translation efficiency was weaker ($r_s$ = –0.12, p<10$^{-11}$).

To test this change in relationship, we examined the protein abundance of specific genes. The transcripts of *gnu*, *ana2*, and *veli* were highly bound by ME31B at 0–1 hr. Consistent with low translation, the encoded proteins were either constant (ANA2, VELI) or decreased (GNU) during the first three hours of development (*Figure 7—figure supplement 1*). In contrast, *smg* mRNA had one of the lowest ME31B binding values in the transcriptome, and SMG protein levels rapidly increased over the first three hours of development (*Figure 7—figure supplement 1*). Thus, high ME31B binding appears to repress translation in early embryos, but, by the end of the MZT, ME31B binding has little effect on translational efficiency.

We next asked whether ME31B binding influenced mRNA decay during this same developmental window. For each time point, we compared ME31B binding to the fold change in mRNA abundance over the next hour of development. Unlike the significant negative relationship between ME31B binding and translation in the 0–1 hr embryo, ME31B binding did not correlate with mRNA decay at this time point (*Figure 7B*; $r_s$ = –0.02, p=0.22). In contrast, ME31B binding was negatively correlated with mRNA stability at all of the later time points: 1–2 to 2–3 hr ($r_s$ = –0.55, p<10$^{-15}$), 2–3 to 3–4 hr ($r_s$ = –0.47, p<10$^{-15}$) and 3–4 to 4–5 hr ($r_s$ = –0.24, p<10$^{-15}$).

To assess this observation with specific transcripts, we used RT-qPCR to monitor *gnu*, *ana2*, and *veli* transcript levels across early development. All three transcripts had relatively high ME31B binding at 1–2 hr, and the abundance of all three decreased between the 1–2 and 2–3 hr time points (*Figure 7—figure supplement 1*). Similarly, although the binding of ME31B to *smg* mRNA was low at 0–1 hr, it increased over the next two hours of embryogenesis. Consistent with our genome-wide analysis, the maximal binding of *smg* by ME31B preceded the sharpest drop in its levels (*Figure 6—figure supplement 1*; *Figure 7—figure supplement 1*). Together, these results indicate that ME31B represses gene expression at all time points, although in different ways. At the beginning of embryogenesis, ME31B binding likely reduces translation but, by the end of the MZT, high ME31B binding is associated with mRNA destabilization.

Finally, we hypothesized that the decreases in ME31B, TRAL, and/or Cup abundance were important for the switch in ME31B function. We thus examined the relationship between ME31B binding and gene regulation in *png*$^{50}$ mutants. As in wild-type embryos, ME31B binding negatively correlated with translational efficiency in *png*$^{50}$ mutant embryos (*Figure 7C*; $r_s$ = –0.24, p<10$^{-15}$). In contrast, although in wild-type embryos hundreds of transcripts were degraded between 0–1 hr and 2–3 hr (*Figure 7D*), in *png*$^{50}$ mutant embryos the RNA abundances between these two time points were nearly identical ($r_s$ = 0.96 vs. $r_s$ = 0.66 in wild-type embryos), and we observed no evidence of any mRNA degradation (*Figure 7E*).

## Discussion

Here we have shown that during the early *Drosophila* MZT, ME31B exists in complexes that also contain eIF4E, Cup, TRAL, and PABP. This complex is likely to be specific to oocytes and early embryos where it represses the translation of many transcripts (*Figure 7F*). The first five hours of embryogenesis coincide with three important changes for the Cup–TRAL–ME31B-containing complex: a sharp reduction in the levels of Cup, TRAL, and ME31B protein and transcripts; a change in the transcripts bound by ME31B; and a change in the impact of ME31B-mediated regulation from repressing translation to destabilizing mRNA, which occurs most likely as the result of repressing translation in the context of robust mRNA decay.

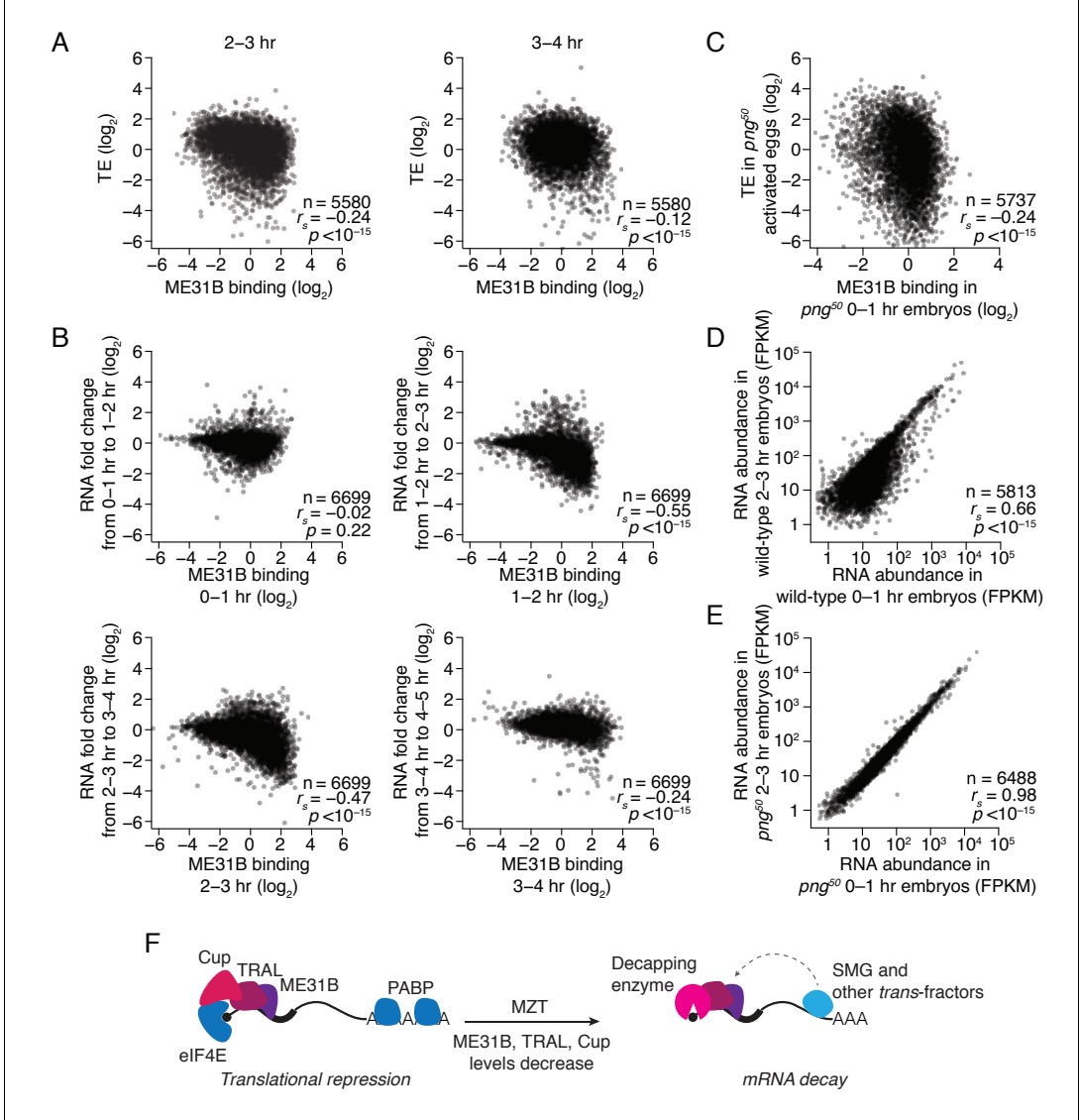

**Figure 7.** The maternal-to-zygotic transition marks an alteration in the post-transcriptional regulation associated with ME31B binding. (A) Comparison of ME31B binding and translational efficiency. Plotted are the ME31B binding and translation efficiency values for each gene expressed above cut-off in 2–3 and 3–4 hr embryos. (B) Comparison of ME31B binding and RNA stability. At each of the indicated time points, the RNA fold change over the next hour was determined. For instance, the RNA fold change between 1–2 hr and 0–1 hr was calculated and compared with ME31B binding at 0–1 hr. Plotted are the ME31B binding and fold-change values for each of the indicated time points. (C) Comparison of ME31B binding and translational efficiency in *png^{50}* embryos. Plotted are the ME31B binding and translational efficiency values for each gene expressed above cut-off in 0–1 hr *eGFP-me31b; png^{50}* embryos (ME31B binding) and 0–1 hr *png^{50}* activated eggs (translational efficiency). (D) Comparison of RNA abundance in wild-type embryos. Plotted are the RNA abundances for each gene in 0–1 hr and 2–3 hr wild-type embryos. (E) Comparison of RNA abundance in *png^{50}* embryos. Plotted are the RNA abundances for each gene in 0–1 hr and 2–3 hr *png^{50}* embryos. (F) Model for the role of ME31B during early development. Prior to the MZT, ME31B, TRAL, and Cup are highly associated with eIF4E and PABP, and ME31B binding represses translation of many transcripts. During the MZT, protein and mRNA levels of Cup, TRAL, and ME31B decrease, and the binding of ME31B now requires recruitment via *trans*-factors, such as SMG. As indicated by the dotted line, this recruitment is likely indirect presumably relying upon interactions between *trans*-factors and the CCR4-NOT deadenylase complex and subsequent binding of ME31B and CNOT1. By the end of MZT, ME31B is no longer associated with PABP, and it promotes mRNA decay likely by recruitment of the decapping enzyme (through TRAL, EDC3, and/or HPat).

DOI: https://doi.org/10.7554/eLife.27891.018

The following figure supplement is available for figure 7:

**Figure supplement 1.** Regulation of ME31B targets during the MZT.

DOI: https://doi.org/10.7554/eLife.27891.019

The changes in Cup, TRAL, and ME31B protein abundance may explain the transcriptome-wide alterations in ME31B binding during the MZT. In the early embryo, these proteins are highly abundant, and a recent study estimated their concentrations as 4, 15, and 7 µM, respectively (*Götze et al., 2017*). Their direct, protein-protein interactions appear sufficient to drive assembly on 5′ caps. At this stage, although *trans*-factors or additional mRNA sequences likely stabilize the complex on some transcripts, they are dispensable for its formation. We propose that the reductions in Cup, TRAL, and ME31B levels lead to a different regime where *trans*-factors are required for stable ME31B binding, as is seen elsewhere (*Chen et al., 2014b*; *Mathys et al., 2014*; *Rouya et al., 2014*). Of the three proteins, the loss of Cup has a particularly dramatic effect because stable interactions between the 5′ cap and ME31B likely require the Cup–eIF4E interaction.

Although Cup, TRAL, and ME31B have been implicated in regulating a small number of transcripts in oogenesis and embryogenesis (*Wilhelm et al., 2003*, *2005*; *Clouse et al., 2008*; *Broyer et al., 2017*), our data indicate that they play a much broader role. ME31B binds transcripts from nearly all of the maternally deposited genes. Its binding negatively correlates with translation in the early embryo, and we speculate that this relationship is a causative one. How, then, might ME31B repress translation in the early embryo? Orthologs of ME31B have been found to inhibit both translation initiation and elongation (*Coller and Parker, 2005*; *Nissan et al., 2010*; *Sweet et al., 2012*; *Radhakrishnan et al., 2016*). However, given the high abundance of Cup, which blocks the eIF4E–eIF4G interaction (*Nelson et al., 2004*; *Kinkelin et al., 2012*), we hypothesize that, at this time point, ME31B predominantly represses translation initiation by Cup-mediated disruption of the eIF4F complex.

Within a few hours of egg laying, the impact of ME31B upon its targets appears to switch from translational repression to mRNA destabilization. Given the tight relationship between translational repression and mRNA decay seen in most contexts (*Radhakrishnan and Green, 2016*), one possible explanation is that at later time points repressed transcripts are quickly degraded and fail to persist long enough to give a translational repression 'signal'. Thus a critical issue may be to understand why ME31B binding fails to stimulate mRNA decapping in the early embryo. There are two, non-mutually exclusive possibilities. In the first, the decapping enzyme (or other related proteins) is inactive or expressed only at low levels in the early embryo. Indeed, such a model was originally proposed when the decoupling of deadenylation and decapping in *Xenopus* oocytes was first observed (*Gillian-Daniel et al., 1998*; *Zhang et al., 1999*). Consistent with this view, one study found that Dcp1a levels are low in mouse oogenesis, only increasing during meiosis (*Flemr et al., 2010*).

An alternative model invokes factors that stabilize ME31B-bound transcripts in the early embryo so that, during the MZT, their inactivation enables clearance of maternal transcripts. One obvious candidate is Cup. Support for such a role for Cup comes from several observations. When Cup was tethered to reporter transcripts in S2 cells, it simultaneously repressed translation and stabilized deadenylated intermediates, possibly by stabilizing eIF4E binding and so sterically interfering with access of the decapping enzyme to the 5′ cap (*Igreja and Izaurralde, 2011*). Moreover, in several organisms, including *Drosophila*, loss of ME31B and Cup have, intriguingly, been linked with decreased, rather than increased, levels of targets (*Mair et al., 2006*; *Boag et al., 2008*; *Broyer et al., 2017*). We thus favor a model whereby ME31B initially represses translation through Cup, and then, only after Cup is destroyed, the major impact of ME31B is at the level of mRNA decay.

One implication of our model is that the reduction in Cup levels is required for the clearance of maternally deposited transcripts, and that the timing of this elimination is critical for embryogenesis. Consistent with such a model, in $png^{50}$ mutants, where Cup, TRAL, and ME31B persist, maternal transcripts were not destroyed, although ME31B binding still appeared to repress translation (*Figure 7*). However, loss of PNG activity at the oocyte-to-embryo transition has pleiotropic effects, including a failure to induce SMG expression (*Tadros et al., 2007*; *Kronja et al., 2014b*), and so this result, while supportive, does not provide definitive evidence. In the future, disentangling the reductions in Cup levels and the ability of ME31B to repress translation after the MZT will be important.

Cup itself is restricted to *Drosophilidae*, but a central role for ME31B in embryogenesis may be evolutionarily conserved. Related complexes have been found in *X. laevis*, *C. elegans* and *Plasmodium* (*Ladomery et al., 1997*; *Boag et al., 2008*; *Mair et al., 2010*). Regulation of ME31B protein and RNA levels is also conserved in other invertebrates as well as in vertebrates. We, therefore, re-examined previously published datasets from early zebrafish development (*Bazzini et al., 2012*) and

found that transcripts encoding zebrafish ME31B (DDX6) decreased during the MZT (*Figure 5—figure supplement 1*). The clearance of DDX6 transcripts was dependent on Dicer (*Figure 5—figure supplement 1*), suggesting that, in zebrafish, regulation of DDX6 mRNA is likely to be mediated by miR-430, the major miRNA at that developmental stage (*Giraldez et al., 2006*). Similarly, previous work in *C. elegans* demonstrated that the protein abundances of ME31B and TRAL orthologs decrease significantly at the 4 cell stage, which coincides with the activation of zygotic transcription (*Boag et al., 2005*). Likewise, *Xenopus* ME31B protein decreases during the MZT (*Ladomery et al., 1997*; *Peshkin et al., 2015*).

Finally, although less dramatic than the shift seen for ME31B-containing complexes, those containing PABP (but not ME31B) also change during the MZT. In the early embryo, transcripts in this complex bind more PABP with increasing tail length. However, this relationship is lost by the end of the MZT. These results echo our observations that, in most cell types, including *S. cerevisiae*, *Drosophila* S2 cells, and human HEK293 cells, poly(A)-tail length has little influence on PABP binding (Rissland *et al.*, in revision). Although the direct mechanisms underlying this shift are unknown, we speculate that the changes in how PABP binds poly(A) tails is a proximal cause for the decoupling of poly(A)-tail length and translational efficiency at the end of the MZT (*Eichhorn et al., 2016*).

## Materials and methods

### Cell lines
S2 cells were purchased from Life Technologies, verified upon shipment to be mycoplasma free. Cells were maintained in ExpressFive SFM media, supplemented with 10% FBS (heat-inactivated) and 20 mM L-Glutamine, at 28°C.

### Drosophila fly stocks
Fly stocks were maintained in a 25°C room with 65% humidity according to standard protocols.

### Transient transfections
S2 cells were transfected using Effectene transfection reagent according to a modified version of the manufacturer's protocol. Briefly, $2 \times 10^6$ cells were seeded in a 6-well plate, and after 24 hr, the cells were transfected with 1 µg of eGFP-Me31B or FLAG-eGFP plasmid. After 48 hr post-transfection, cells were harvested, lysed in Buffer A (100 mM KCl, 0.1 mM EDTA, 20 mM HEPES-KOH pH 7.6, 0.4% NP-40, 10% glycerol, 1 mM DTT, complete mini EDTA-free protease inhibitors, SUPERase-In RNase inhibitors), and clarified at 14,000 rpm, 4°C for 5 min. The supernatant was then immediately used for immunoprecipitations, western blot analysis and/or RNA extraction.

### Isolation of embryos and stage 14 oocytes
Embryos were collected at various time points post-egg laying, dechorionated with bleach, and washed with 0.1% Triton X-100. Embryos were then homogenized in lysis Buffer B (150 mM KCl, 20 mM HEPES-KOH pH 7.4, 1 mM MgCl$_2$, 1 mM DTT, complete mini EDTA-free protease inhibitors), and were clarified at 15,000 rpm, 4°C for 15 min. The supernatant was stored at –80°C. Stage 14 oocytes were isolated from a large-scale *Drosophila* culture established with $w^{1118}$ flies, and were homogenized in lysis buffer B (with protease inhibitor cocktail (BioShop, Canada) and additional freshly added protease inhibitors [100 µM Leupeptin, 100 µM Chymostatin, 4 mM Benzamidine HCl, 3 µM Pepstatin; Sigma] and SUPERase-In RNase inhibitors). The homogenized lysates were clarified at 15,000 rpm, 4°C for 15 min, and the supernatant was stored at –80°C.

### Immunoprecipitations
The anti-PABP (Fab1: D032; Fab 2: D035), anti-eIF4G (P190), and C1 Fabs were generated and purified as previously described (*Na et al., 2016*). Prior to the IP, FLAG-tagged anti-PABP, anti-eIF4G or control C1 Fabs were conjugated to anti-FLAG beads overnight at 4°C. For eGFP-ME31B IP, anti-GFP antibodies were conjugated to protein G beads overnight at 4°C. For RIPs, the beads were also blocked with salmon ssDNA. Lysates were diluted to 1.1 mg/ml, and then incubated with conjugated beads for 2–3 hr, rotating at 4°C. Beads were washed six times with the appropriate lysis buffer. In the case of immunoprecipitations from embryo lysates, lysis Buffer B was supplemented with 0.1%

Triton X-100 and SUPERase-In RNase inhibitors for the washes. For RNA extraction, the beads were resuspended in TRI-Reagent. For western blot analysis, the beads were boiled in LDS sample buffer and reducing agent. For RNase A treatment, the beads were washed two times after immunoprecipitation and incubated with or without RNase A for 30 min at 4°C. The beads were then subsequently washed four times and boiled in LDS sample buffer and reducing agent.

For the PABP and eIF4G IPs, 1% input and 3.5% IP were loaded onto a SDS-PAGE gel and probed for Cup, TRAL, ME31B, eIF4E, and HPat. To probe for PABP and eIF4G, 0.25% input and 0.88% IP were loaded. To probe for EDC3, 3% input and 10.5% IP were loaded. For the eGFP-ME31B IPs, 1% input and 3.5% IP were loaded onto a SDS-PAGE gel and probed for Cup, TRAL, ME31B, eIF4E, and HPat. To probe for PABP and eIF4G, 0.05% input and 0.18% IP were loaded. To probe for EDC3, 3% input and 10.5% IP were loaded. When protein abundance changes during the time course were probed, 5 ng of C1 Fab were added to constant amounts of lysates, as determined by Bradford assay.

## m7GDP cap-analog pull down

Lysates were incubated with blank agarose beads for 10 min at 4°C to pre-clear the lysate. The beads were removed by centrifugation, and the supernatant was incubated with either blank agarose beads or m7GDP-conjugated agarose beads for 3 hr at 4°C. Beads were washed three times with the appropriate lysis buffer and resuspended in LDS sample buffer and reducing agent for western blot analysis. Thereafter, 1% input and 10% pull down were loaded onto a polyacrylamide gel and probed for all the indicated proteins.

## Tandem mass spectrometry

Immunoprecipitates were sent to SPARC BioCentre (SickKids) for LC/MS/MS analysis.

## RNA sequencing

RNA was extracted from immunoprecipitates and input lysate using TRI-reagent. Prior to isopropanol precipitation, 600 ng of *C. elegans* RNA was spiked into samples for normalization purposes. To assess the enrichment of the RNA-immunoprecipitation, a fraction of the RNA was treated with DNase and used for RT-qPCR to ensure enrichment of *Act5C* transcripts. After verifying the quality of the RNA-immunoprecipitation, the RNA was subjected to Ribo-Zero Gold rRNA depletion according to the manufacturer's protocol. Libraries were then generated using Illumina's TruSeq stranded mRNA library preparation kit according to the manufacturer's protocol and sequenced at The Center for Applied Genetics (SickKids).

## Computational analyses

Libraries were pooled and sequenced on an Illumina HiSeq 2500 by The Centre for Applied Genomics at The Hospital for Sick Children. 50 base-pair single-end reads were demultiplexed and converted to FASTQ format using bcl2fastq2 v2.17 (Illumina). Library quality was inspected using FastQC v0.11.5 (http://www.bioinformatics.babraham.ac.uk/projects/fastqc/). Reads were trimmed for quality and clipped for Illumina adaptors using TrimmomaticSE version 0.36 (*Bolger et al., 2014*). Surviving reads were mapped by STAR 2.5.2a (*Dobin et al., 2013*) to the *D. melanogaster* genome or combined *D. melanogaster* (dm6) and *C. elegans* (ce10) genomes obtained from UCSC on 7 August 2016. Genes were quantified using Cufflinks 2.2.1 (*Trapnell et al., 2010*). Downstream analyses were then performed with R version 3.1.2, using in-house scripts. To normalize for global changes in mRNA abundance, input samples were normalized to the total number of reads mapping to the *C. elegans* genome. Binding for each gene was calculated by dividing the IP FPKM by the unnormalized input FPKM. When calculating binding, all the genes were filtered such that only genes with greater than 0.5 FPKM across all time points were included in the analysis. For input calculations, because we expected greater changes in RNA level due to developmental events, we reduced the stringency of the filter such that 0.5 FPKM of a gene at a single time point was sufficient for inclusion in downstream analysis. To calculate enrichment of transcripts in PABP immunoprecipitations over ME31B immunoprecipitations, PABP binding for each gene was divided by its ME31B binding.

## Statistical analysis

Two biological replicates were performed for the PABP and ME31B RIP-seq experiments, using lysates harvested on different days. ME31B and PAPB immunoprecipitations were performed from different lysates. At each time point, the input and immunoprecipitation FPKM values were well-correlated between replicates ($r_s > 0.75$ for all replicate comparisons). For simplicity, one replicate is presented. High-throughput sequencing data described in this paper are available from the GEO: GSE83616 and, for the data prepared in this paper, GSE98106.

## Acknowledgements

We thank Dr. Andrew Spence, Dr. Julie Claycomb and members of the Rissland, Claycomb, and Smibert labs for insightful questions and stimulating conversations. We especially thank Dr. Phillip D Zamore for his thought-provoking feedback and advice. We are also grateful to Dr. Elisa Izaurralde, Dr. José Sierra, Dr. Silke Dorner, Dr. Elisabeth Knust, Dr. Akira Nakamura, Dr. Terry Orr-Weaver, and Dr. Jordan Raff for antibodies. This work was funded by NSERC Discovery Grants (to OSR and CAS), an Operating Grant from the CIHR (to HDL, MOP-14409) a University of Toronto Open Fellowship (to MW), an Ontario Graduate Scholarship award (to AL), and an NSERC CGS-M award (to AL).

## Additional information

### Funding

| Funder | Author |
|---|---|
| Natural Sciences and Engineering Research Council of Canada | Craig A Smibert<br>Olivia S Rissland |
| Canadian Institutes of Health Research | Howard D Lipshitz |

The funders had no role in study design, data collection and interpretation, or the decision to submit the work for publication.

### Author contributions

Miranda Wang, Conceptualization, Formal analysis, Investigation, Writing—original draft, Writing—review and editing; Michael Ly, Conceptualization, Formal analysis, Investigation, Methodology, Writing—original draft, Writing—review and editing; Andrew Lugowski, Investigation, Methodology, Writing—original draft, Writing—review and editing; John D Laver, Conceptualization, Formal analysis, Writing—original draft, Writing—review and editing; Howard D Lipshitz, Craig A Smibert, Conceptualization, Writing—original draft, Writing—review and editing; Olivia S Rissland, Conceptualization, Resources, Formal analysis, Supervision, Funding acquisition, Investigation, Methodology, Writing—original draft, Project administration

### Author ORCIDs

Olivia S Rissland, http://orcid.org/0000-0002-2619-6019

### Decision letter and Author response

Decision letter https://doi.org/10.7554/eLife.27891.025
Author response https://doi.org/10.7554/eLife.27891.026

## Additional files

### Supplementary files

• Transparent reporting form
DOI: https://doi.org/10.7554/eLife.27891.020

## Major datasets

The following dataset was generated:

| Author(s) | Year | Dataset title | Dataset URL | Database, license, and accessibility information |
|---|---|---|---|---|
| Wang M, Ly M, Lugowski A, Laver JD, Lipshitz HD, Smibert CA, and Rissland OS | 2017 | ME31B globally represses maternal mRNAs by two distinct mechanisms during the Drosophila maternal-to-zygotic transition | https://www.ncbi.nlm.nih.gov/geo/query/acc.cgi?acc=GSE98106 | Publicly available at the NCBI Gene Expression Omnibus (accession no: GSE98106) |

The following previously published dataset was used:

| Author(s) | Year | Dataset title | Dataset URL | Database, license, and accessibility information |
|---|---|---|---|---|
| Eichhorn SW, Subtelny AO, Kronja I, Orr- Weaver TL, Bartel DP | 2016 | mRNA Poly(A)-tail Changes Specified by Deadenylation Broadly Reshape Translation in Drosophila Oocytes and Early Embryos | https://www.ncbi.nlm.nih.gov/geo/query/acc.cgi?acc=GSE83616 | Publicly available at the NCBI Gene Expression Omnibus (accession no: GSE83616) |

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
