## [Decision Letter]

Thank you for submitting your article "ME31B globally represses maternal mRNAs by two distinct mechanisms during the *Drosophila* maternal-to-zygotic transition" for consideration by *eLife*. Your article has been favorably evaluated by K VijayRaghavan (Senior Editor), a Reviewing Editor, and three reviewers. The following individual involved in review of your submission has agreed to reveal his identity: Jeff Coller (Reviewer #1).

The reviewers have discussed the reviews with one another and the Reviewing Editor has drafted this letter to help us evaluate the feasibility of revisions on this manuscript.

We have received comments from three experts in the field all of whom found this study reporting on the changing nature of mRNP composition over the course of the *Drosophila* MZT to be of broad interest and to reveal some novel insights into developmental regulation of gene expression. Despite broad enthusiasm there are a number of critical issues that require work in order for this study to be appropriate for publication at *eLife*.

1) The first critical issue concerns the statistical approaches used in assessing correlations among diverse normalized datasets, as detailed by reviewer #2. The correlations that the authors report are at the heart of the manuscript and must be critically evaluated using alternative statistical approaches that avoid the "common denominator input" that can lead to spurious correlations. This may be as simple as comparing the correlations of the RIP RPKMs for Me31b and Pabp directly (without "input" division). If they are binding the same targets, these RPKMs should be positively correlated.

2) The second point, assuming that the correlations stand up to the new statistical analysis, is that the conclusions of the study rely very strongly on correlative data. The conclusions of the manuscript, that changes in mRNP composition due to changing protein levels lead to changes in the form of regulation, would be substantially bolstered by validating changes in Me31b and pAbp binding, and mRNA abundance by qPCR for other mRNA (not just *Act5c*), and evidence for translational repression by western blot analysis for several proteins.

3) The conclusions would also be strengthened by repeating the high throughput analyses in the *png Drosophila* background. The prediction is that the observed changes in "mechanism" of regulation would no longer be observed, since the critical proteins are no longer down regulated.

4) Finally, even if the correlations hold, and the *png* experiment yields the predicted result, the overall discussion of "mechanism" needs to be substantially rewritten. The manuscript argues that the change in mRNP composition leads to a change in "mechanism" of regulation, but fails to acknowledge that what might have changed during the MZT is simply the induction of the mRNA degradation pathway which then erases the signature of translational control during the late phases of the MZT. This more measured discussion of the data would be easily reconciled with considerable literature in this area and should be entertained as a possibility. While it is true that the authors following changes in protein levels of ME31B, Cup and Tral, what they follow is the activities of mRNAs that associate with these proteins. It is not clear that there is not enough protein even at the end of the MZT to achieve the same type and level of regulation.

*Reviewer #1:*

In the manuscript "ME31B globally represses maternal mRNAs by two distinct mechanisms during the *Drosophila* maternal-to-zygotic transition", Rissland's group performs immunoprecipitation followed by Western blotting or RNA-Seq to demonstrate that, in the early embryo, ME31B forms complex with Cup, TRAL to repress translation. During MZT, this complex dissociates and ME31B promotes mRNA degradation.

This is an interesting study to explore how maternal mRNAs are repressed and cleared by distinct mechanisms which are conducted by ME31B. There are some concerns for the authors to strengthen the importance of this study:

There are two main concepts of this manuscript. First, in the early embryo, ME31B-Cup-TRAL complex represses translation of target RNAs. The authors show the negative correlation between ME31B binding and TE in 0-1 hr embryo, while the correlation becomes weaker at 3-4 hr embryo. Second, during MZT, ME31B-Cup-TRAL complex dissociate and the association of ME31B with transcripts triggers mRNA degradation. It will be more convincing if the authors can show the translation of some ME31B targets are actually repressed in early embryo by performing Western blotting and mRNA are degraded during MZT by performing Northern blotting.

RIP experiments indicate that most of transcripts expressed during embryogenesis are bound by ME31B or PABP. In Figure 4, the authors show that, ME31B bound transcripts at 0-1 hr embryo is weakly correlated with those at MZT (2-3, 3-4 and 4-5 hr). However, the percent *Act5C* IP'ed by ME31B decreases from 42,000 fold to 4 fold, and the percent *Act5C* IP'ed by control IgG increases in the later time points (By comparing Figure 2 and Figure 6—figure supplement 1). Consider both the level of ME31B (Figure 4) and the enrichment of benchmark *Act5C* (Figure 6—figure supplement 1) dramatically decrease after 2-3 hr of developmental stage, how many transcripts are really bound by ME31B in the later time courses? Can authors provide RIP-Seq data from control antibody or show the enrichment of ME31B IP relative to control antibody for few more examples (besides of *Act5C*) over different time points. This would be an important control to show the enrichment of the vast majority of expressed genes in ME31B pull-down is not due to background noises.

The authors show that, mRNA and protein levels of ME31B, Cup and TRAL decrease during MZT and PNG kinase is required to regulate ME31B, Cup and TRAL expression. In *png50* mutant embryo, ME31B, Cup and TRAL failed to decline and still formed the complex during MZT. How does *png50* mutant impact mRNA levels during MZT? If the authors can show that, in *png50* mutant, ME31B-Cup-TRAL complex still repress translation and mRNA can't be further degraded by performing Western and Northern blotting for some ME31B targets, it would be a solid evidence to support the proposed model.

Some data and experimental procedures need to be clarified:

The authors mention that, Ypsilon schachtel (YPS) co-immunoprecipitated with PABP. However, this data is missing in Figure 1.

It would be better to use "association" or "enrichment" instead of "occupancy" for the RIP-Seq results.

In Figure 1, there's no difference of ME31B levels between the input of S2 cells and 0-1 hr embryo. However, in Figure 4, ME31B levels in 0-1 hr embryo is 15-20 fold more than it in S2 cells. Authors need to explain the inconsistence of these two data.

*Reviewer #2:*

Wang and colleagues here report on the changing nature of mRNP composition over the course of the *Drosophila* MZT. They provide evidence that the ME31B protein (homolog to human *ddx6* and yeast *dhh1*) associates with PABP, Cup, and Tral in early embryos, and that this mRNP complex is remodeled throughout development. They performed RIP-seq analysis of PABP and ME31B protein and report on the changing nature of several correlations between occupancy of these proteins and other features of the transcriptome (polyA length, translation efficiency, and changes in RNA levels. They put forth that their data fit with a model in which ME31B mRNP remodeling changes the function of this RBP from a translational repressor to that of an mRNA decay enhancer. The coIP experiments and western blot analyses are solid and the evidence for mRNP remodeling is strong. The results are also very interesting and timely in that they address how the role of this protein might change during the MZT. However, I have significant doubts about the statistical approaches used in assessing correlations among diverse normalized datasets. These doubts could potentially be eased through alternative statistical analyses in consultation with a statistician and additional experiments. I also have other concerns about the nature or RIP-seq, as discussed below:

1) RIP-seq analysis has gone out of fashion, so to speak, largely because of concerns that interactions between proteins and RNA become jumbled during immunoprecipitation (i.e. non-specific noise). RBPs could release from one mRNA and bind another. Some mRNAs not bound by the protein targeted could be precipitated through base-pairing interactions with bonafide bound mRNAs, etc. This is why ClIP has become a more standard approach, allowing more stringent digestion of RNA and washes, etc. Thus the authors’ RIP-seq results may be subject to this problem. This is especially troublesome since ME31B appears to be bound somewhat ubiquitously. One way to address this would be to mix *C. elegans* extracts (or those from some other species) into their IPs (from eGFP tagged ME31B. This control would allow the authors to test whether their IP is pulling out somewhat random aggregates. Spiking in *C. elegans* RNA only would allow the authors to investigate possible base-pairing creating background noise.

2) Correlation analyses of the IP datasets are fraught with potential statistical problems. First, RPKM data are compositional (they add up to a constant). Compositional data, being relative levels akin to% 's, have different properties from absolute data and need to be handled differently for correlation analysis (see http://journals.plos.org/ploscompbiol/article?id=10.1371/journal.pcbi.1004075#pcbi.1004075.ref004 for one example). In particular, this can lead to the appearance of spurious correlations. The authors should re-evaluate their data using methods developed for compositional data analysis (e.g. see http://www.compositionaldata.com), or (preferably) consult a biostatistician on the issue.

3) The authors introduce a metric they call "occupancy". This divides the FPKM of IP RNA by that of input RNA. This has an unfortunate propensity to produce potentially spurious correlations. This is well documented in the statistics literature, but not always appreciated by biologists. To explain this, if one draws two sets of random data from a normal distribution (rnorm in R) and plots them against each other, there's no correlation. However, if one then takes a third random dataset and divides each of the first two by it, a positive correlation is seen. The reason behind this is that both of the datasets now have a common denominator, and that common denominator drives the correlation. In this case, the authors data are roughly log-normal, so the following R-code would show the spurious relationship:x1 = rnorm(10000, 5, 1) # analogous to ME31B RIP FPKMsx2 = rnorm(10000, 5, 1) # analogous to PABP RIP FPKMsy = rnorm(10000, 5, 1) # analogous to Input FPKMs.cor.test(x1, x2, method=c("spearman")) # will return no correlationplot(x1,x2) # same visuallycor.test(x1-y, x2-y, method=c("spearman")) # will return a big positive correlation of rho = 0.48, P < 2.2e^-16.plot(x1-y,x2-y) # same visually

Here I used x1-y to represent the Log(X1/Y) because the data are log-normal log(X1) – log(Y) = log(X1/Y).

So the result in Figure 2 (positive correlation between PABP occupancy and ME31B occupancy) could be spurious – coming from the common denominator input for both occupancies. The fact that this is not seen in Figure 2 suggests there are differences between the Embryo and S2 cell datasets, but it's very hard to interpret with this confounding denominator issue. Figure 6 shows decreased correlations over time, but these could be largely driven by lower correlations between input RNA-seq FPKMs (denominators) over time. Many results in the paper rely on this kind of analysis (occupancy vs. something else, where both are divided by mRNA levels (e.g. TE)). This potentially affects Figure 2, Figure 3, Figure 6, and 7. A biostatistician should be consulted to help sort out the best way to handle this. In addition, it would help to validate the RIP data (assuming it's clean see point 1) via northern blot or qRT-PCR for more genes than *Act5C*.

4) Somewhat / potentially minor question and comment. The IP data in Figure 1 are interesting in that there is a second isoform of eIF4G that is abundant in S2 cells and present in the embryo lysate, but does not IP. What is this isoform and why is it not part of the eIF4G complex with PABP and eIF4E in embryos?

*Reviewer #3:*

Wang et al. here explore the role of the RNA binding protein ME31B during the Zebrafish maternal to zygotic transition (MZT). They show that in early embryos, ME31B, along with Cup and Tral, associate with PABP, but not eIF4G, suggesting the presence of a complex that inhibits the closed loop interaction (consistent with much earlier work). Over the time course of the MZT, they see that the levels of ME31B, Cup and Tral substantially decline, and their pull-down by IP with PABP similarly declines. With these ideas in place, they explore by RIP-seq, RNA-seq and ribosome profiling how these changes in protein amounts and interactions correlate with mRNA levels and translational efficiency (and gathering some already published data from the literature). First, they find by RIP-seq that in early embryos PABP and ME31B bind similar sets of mRNAs (or at least enrich mRNAs in a correlated way), but that these binding interactions become de-correlated during development. They show that at early time points, there is strong (negative) correlation between ME31B binding and TE (and polyA tail length), while this correlation disappears beyond these early time points. They show that changes in mRNA levels do not correlate with ME31B binding at early time points, but that this (negative) correlation increases at the later time points. Thus these trends for changes in mRNA levels and for translational efficiency are seemingly mirror images of one another. These data together are interesting and well presented and the coincidence in timing of changes in protein levels and in effects on output (mRNA and TE) is intriguing. These data also fit nicely with earlier studies characterizing the correlations between polyA tail length and TE during this same developmental period (an early correlation that is seemingly lost at later time points). However, importantly, these relationships are all at this stage simply correlative, and are not established as causal. The authors do explore the effects of two different factors on the trends they observe, and they nicely show the PNG kinase is critical for the loss of the ME31B, Cup, Tral proteins, but that SMG is not. But they do not connect directly the proteins that are the focus of the study, ME31B, Cup and Tral, to the phenomena that they describe through genetic approaches, as would be required to establish causality.

As a final but substantive point, I find the conclusions that the authors reach to be very much overstated for the type of data they have gathered. What is presented in this study, and in many previous studies of this nature, is a set of correlations that change during this time course of development. There are interesting events being explored and the data gathered are impressive. But the observation of such correlations does not permit strong statements of mechanism that depend on mutational analyses and careful ordering of events. As acknowledged by the authors in their Introduction, there are *many* changes happening during the MZT, including the activation of mRNA decay pathways that are silenced in the early embryo. Given that translation and mRNA decay are tightly coupled in so many systems, it seems at a minimum that these dependencies would be discussed as a possible (and indeed likely) explanation for what is observed in this system. And, since this possibility can't be eliminated by the analysis here, then it should be offered throughout as an alternative to the one preferred by the authors which is that the "mechanism" has changed from one of translational repression to one of mRNA decay during the MZT. Stated more clearly, if the decay machinery wasn't expressed early in the time course, then all that will be observed is translational repression, but if they are coupled later in the time course (as in much of biology), then the mRNA decay that happens will eliminate the signature of the translational repression. In the absence of mutations that separate these events, these alternative explanations must be offered as equally likely. In terms of simplicity, it seems easier to argue that translational repression is common throughout development, but that the increasing levels of decay machinery is a critical change that triggers the apparent change in "mechanism". The problem for the authors is that they see changes in protein levels (ME31B, Cup and Tral), but fundamentally, despite these changes, they are following the activities of mRNAs that still associate with these proteins.

---

## [Author Response]

We have received comments from three experts in the field all of whom found this study reporting on the changing nature of mRNP composition over the course of the Drosophila MZT to be of broad interest and to reveal some novel insights into developmental regulation of gene expression. Despite broad enthusiasm there are a number of critical issues that require work in order for this study to be appropriate for publication at eLife.1) The first critical issue concerns the statistical approaches used in assessing correlations among diverse normalized datasets, as detailed by reviewer #2. The correlations that the authors report are at the heart of the manuscript and must be critically evaluated using alternative statistical approaches that avoid the "common denominator input" that can lead to spurious correlations. This may be as simple as comparing the correlations of the RIP RPKMs for Me31b and Pabp directly (without "input" division). If they are binding the same targets, these RPKMs should be positively correlated.

We thank the reviewers, especially reviewer 2, for raising this issue. The reviewers are correct that dividing by a common denominator can introduce spurious correlations into datasets. However, this is not the case with our datasets. When we compared the RIP FPKMs for ME31B and PABP (that is, without input normalization), we observed higher correlations than with the input-normalized occupancies. For instance, in 0–1 hr embryos, the correlation between the RIP FPKMs in 0.94, while the correlation between occupancies is 0.62. Thus, we conclude that the division by a common denominator does not drive these correlations, but instead corrects for the impact of RNA abundance on the RIP FPKMs. We have included a more detailed discussion of this analysis in the paper and added the scatter plots to Figure 2—figure supplement 1.

We have also compared our binding values from the RIP-seq to those determined using RIP followed by RT-qPCR for five transcripts that span a range of occupancies. The results of seq versus RT-qPCR based methods are in accordance, and these data have now been included (Figure 2—figure supplement 2; Figure 6—figure supplement 1 and Figure 6—figure supplement 2). These analyses also revealed that even some of the lowliest bound transcripts (such as *smg*) are still enriched in ME31B and PABP immunoprecipitations across this developmental window (see Figure 6—figure supplement 1 and Figure 6—figure supplement 2).

2) The second point, assuming that the correlations stand up to the new statistical analysis, is that the conclusions of the study rely very strongly on correlative data. The conclusions of the manuscript, that changes in mRNP composition due to changing protein levels lead to changes in the form of regulation, would be substantially bolstered by validating changes in Me31b and pAbp binding, and mRNA abundance by qPCR for other mRNA (not just Act5c), and evidence for translational repression by western blot analysis for several proteins.

We very much appreciate this suggestion, and, as anticipated by the reviewers, including additional examples strengthened our results. We identified target genes with (1) high ME31B binding at 0–1 hr, (2) low translational efficiency at 0–1 hr, and (3) antibodies available. These include *GNU, Ana2*, and *Veli*. We analyzed their protein expression and total RNA levels, as well as ME31B/PABP binding, over the first five hours of development in wild-type embryos. Each of these proteins did not increase or, in fact, decreased over the first three hours of embryogenesis, and their corresponding transcript levels decreased at 2–3 hr. Thus, these results are consistent with our model where the impact of ME31B changes from repressing translation to stimulating mRNA decay during the MZT.

As a control, we also analyzed *SMG*, whose transcript has low ME31B binding and high translational efficiency at 0–1 hr. Consistent with our transcriptome-wide analysis, SMG protein increases from 0–1 hr to 1–2 hr. Interestingly, ME31B binding to *smg* also increases over the first three hours of embryogenesis, and, in line with ME31B stimulating mRNA decay later, *smg* transcript levels decrease at 3–4 hr.

We have described these new results in the main text and have included these data in supplemental figures (Figure 6—figure supplement 1 and Figure 6—figure supplement 2; Figure 7—figure supplement 1).

3) The conclusions would also be strengthened by repeating the high throughput analyses in the png Drosophila background. The prediction is that the observed changes in "mechanism" of regulation would no longer be observed, since the critical proteins are no longer down regulated.

Like the reviewers, we had also hypothesized that this might be the case. Accordingly, we have performed RNA sequencing and ME31B RIP-seq in *png* mutant embryos. ME31B occupancy still negatively correlated, albeit weakly, with translational efficiency in 0–1 hr *png* mutant embryos (*r_s_* = –0.25), which is similar to the results we see in wild-type embryos (*r_s_* = –0.29). These data suggest that Cup–Tral–ME31B complex is still able to repress translation.

In contrast, when we compared RNA abundance between 0–1 hr and 2–3 hr embryos in wild-type and *png* mutants, we observed no mRNA decay. In wild-type embryos, many transcripts have been degraded by this point, as has been described by other groups. However, there is no evidence of mRNA destabilization in the *png* mutant embryos at the same stage, an observation that is consistent with our model. We have included these data in Figure 7.

4) Finally, even if the correlations hold, and the png experiment yields the predicted result, the overall discussion of "mechanism" needs to be substantially rewritten. The manuscript argues that the change in mRNP composition leads to a change in "mechanism" of regulation, but fails to acknowledge that what might have changed during the MZT is simply the induction of the mRNA degradation pathway which then erases the signature of translational control during the late phases of the MZT. This more measured discussion of the data would be easily reconciled with considerable literature in this area and should be entertained as a possibility. While it is true that the authors following changes in protein levels of ME31B, Cup and Tral, what they follow is the activities of mRNAs that associate with these proteins. It is not clear that there is not enough protein even at the end of the MZT to achieve the same type and level of regulation.

We thank the reviewers for this general point about our Discussion, and we have rewritten it to address their concerns.

In regards to ME31B proteins levels, we do note that, based on recent estimates of protein concentrations from the Wahle group (Götze, et al. 2017), ME31B is present at 7 μM in the early embryo, and so, even a 10– to 20–fold decrease in abundance during the MZT would still lead to ME31B concentrations in line with those observed in yeast and S2 cells, where its ability to repress gene expression has been well-established.

Below are our responses to specific comments from each reviewer:

Reviewer #1:[…] This is an interesting study to explore how maternal mRNAs are repressed and cleared by distinct mechanisms which are conducted by ME31B. There are some concerns for the authors to strengthen the importance of this study:There are two main concepts of this manuscript. First, in the early embryo, ME31B-Cup-TRAL complex represses translation of target RNAs. The authors show the negative correlation between ME31B binding and TE in 0-1 hr embryo, while the correlation becomes weaker at 3-4 hr embryo. Second, during MZT, ME31B-Cup-TRAL complex dissociate and the association of ME31B with transcripts triggers mRNA degradation. It will be more convincing if the authors can show the translation of some ME31B targets are actually repressed in early embryo by performing Western blotting and mRNA are degraded during MZT by performing Northern blotting.

We thank the reviewer for this suggestion. As described above, we have now included these data in Figure 7—figure supplement 1.

RIP experiments indicate that most of transcripts expressed during embryogenesis are bound by ME31B or PABP. In Figure 4, the authors show that, ME31B bound transcripts at 0-1 hr embryo is weakly correlated with those at MZT (2-3, 3-4 and 4-5 hr). However, the percent Act5C IP'ed by ME31B decreases from 42,000 fold to 4 fold, and the percent Act5C IP'ed by control IgG increases in the later time points (By comparing Figure 2 and Figure 6—figure supplement 1). Consider both the level of ME31B (Figure 4) and the enrichment of benchmark Act5C (Figure 6—figure supplement 1) dramatically decrease after 2-3 hr of developmental stage, how many transcripts are really bound by ME31B in the later time courses? Can authors provide RIP-Seq data from control antibody or show the enrichment of ME31B IP relative to control antibody for few more examples (besides of Act5C) over different time points. This would be an important control to show the enrichment of the vast majority of expressed genes in ME31B pull-down is not due to background noises.

We have now included qPCR data for additional transcripts, including *gnu, ana2, veli*, and *smg*. Importantly, *smg* was in the bottom decile for ME31B binding at 4–5 hrs, but, as measured by RT-qPCR, this transcript was still enriched in ME31B immunoprecipitations relative to control ones. We have included these data in Figure 2—figure supplement 2 and Figure 6—figure supplement 1 and Figure 6—figure supplement 2.

The authors show that, mRNA and protein levels of ME31B, Cup and TRAL decrease during MZT and PNG kinase is required to regulate ME31B, Cup and TRAL expression. In png50 mutant embryo, ME31B, Cup and TRAL failed to decline and still formed the complex during MZT. How does png50 mutant impact mRNA levels during MZT? If the authors can show that, in png50 mutant, ME31B-Cup-TRAL complex still repress translation and mRNA can't be further degraded by performing Western and Northern blotting for some ME31B targets, it would be a solid evidence to support the proposed model.

As described above, we found that ME31B binding negatively correlated with translational efficiency in png50 mutants, but, consistent with the reviewer’s prediction, we observed no mRNA decay in mutant embryos. These data have now been included in Figure 7.

Some data and experimental procedures need to be clarified:The authors mention that, Ypsilon schachtel (YPS) co-immunoprecipitated with PABP. However, this data is missing in Figure 1.

We thank the reviewer for this point, and we have clarified the text.

It would be better to use "association" or "enrichment" instead of "occupancy" for the RIP-Seq results.

We have removed “occupancy” throughout.

In Figure 1, there's no difference of ME31B levels between the input of S2 cells and 0-1 hr embryo. However, in Figure 4, ME31B levels in 0-1 hr embryo is 15-20 fold more than it in S2 cells. Authors need to explain the inconsistence of these two data.

We apologize for any confusion in Figure 1. These two western blots were performed separately (and are two different subfigures) and so the input protein levels cannot be directly compared in this figure.

Reviewer #2:Wang and colleagues here report on the changing nature of mRNP composition over the course of the Drosophila MZT. They provide evidence that the ME31B protein (homolog to human ddx6 and yeast dhh1) associates with PABP, Cup, and Tral in early embryos, and that this mRNP complex is remodeled throughout development. They performed RIP-seq analysis of PABP and ME31B protein and report on the changing nature of several correlations between occupancy of these proteins and other features of the transcriptome (polyA length, translation efficiency, and changes in RNA levels. They put forth that their data fit with a model in which ME31B mRNP remodeling changes the function of this RBP from a translational repressor to that of an mRNA decay enhancer. The coIP experiments and western blot analyses are solid and the evidence for mRNP remodeling is strong. The results are also very interesting and timely in that they address how the role of this protein might change during the MZT. However, I have significant doubts about the statistical approaches used in assessing correlations among diverse normalized datasets. These doubts could potentially be eased through alternative statistical analyses in consultation with a statistician and additional experiments. I also have other concerns about the nature or RIP-seq, as discussed below:1) RIP-seq analysis has gone out of fashion, so to speak, largely because of concerns that interactions between proteins and RNA become jumbled during immunoprecipitation (i.e. non-specific noise). RBPs could release from one mRNA and bind another. Some mRNAs not bound by the protein targeted could be precipitated through base-pairing interactions with bonafide bound mRNAs, etc. This is why ClIP has become a more standard approach, allowing more stringent digestion of RNA and washes, etc. Thus the authors’ RIP-seq results may be subject to this problem. This is especially troublesome since ME31B appears to be bound somewhat ubiquitously. One way to address this would be to mix C. elegans extracts (or those from some other species) into their IPs (from eGFP tagged ME31B. This control would allow the authors to test whether their IP is pulling out somewhat random aggregates. Spiking in C. elegans RNA only would allow the authors to investigate possible base-pairing creating background noise.

As the reviewer notes, RIP-based approaches do include in vitro incubation and washes that might perturb the native occupancies of some RNA-binding proteins. Because CLIP protocols had not yet been developed for *Drosophila* embryos when we began our studies, we modified the RIP protocol to minimize the duration of the in vitro incubation and potential effects of release and rebinding. Furthermore, there are concerns over the inability of UV light to fully penetrate the fly embryo given its large size. Also, the non-quantitative nature of CLIP and PAR-CLIP are a concern in studies such as ours (see Nicholson, et. al 2016). We note that RIPs followed by microarrays, sequencing and Nanostring have been used by multiple groups to identify targets of other RNA binding proteins in *Drosophila* embryos and elsewhere and to determine the effect of these proteins on gene expression (Lebedeva, et al. 2011; Mukherjee, et al. 2011; Laver, et al. 2013; Chen, et al. 2014; Laver, et al. 2015; Nicholson, et al. 2017).

2) Correlation analyses of the IP datasets are fraught with potential statistical problems. First, RPKM data are compositional (they add up to a constant). Compositional data, being relative levels akin to% 's, have different properties from absolute data and need to be handled differently for correlation analysis (see http://journals.plos.org/ploscompbiol/article?id=10.1371/journal.pcbi.1004075#pcbi.1004075.ref004 for one example). In particular, this can lead to the appearance of spurious correlations. The authors should re-evaluate their data using methods developed for compositional data analysis (e.g. see http://www.compositionaldata.com), or (preferably) consult a biostatistician on the issue.

We thank the reviewer for this feedback and have addressed this point as described above.

3) The authors introduce a metric they call "occupancy". This divides the FPKM of IP RNA by that of input RNA. This has an unfortunate propensity to produce potentially spurious correlations. This is well documented in the statistics literature, but not always appreciated by biologists. To explain this, if one draws two sets of random data from a normal distribution (rnorm in R) and plots them against each other, there's no correlation. However, if one then takes a third random dataset and divides each of the first two by it, a positive correlation is seen. The reason behind this is that both of the datasets now have a common denominator, and that common denominator drives the correlation. In this case, the authors data are roughly log-normal, so the following R-code would show the spurious relationship:x1 = rnorm(10000, 5, 1) # analogous to ME31B RIP FPKMsx2 = rnorm(10000, 5, 1) # analogous to PABP RIP FPKMsy = rnorm(10000, 5, 1) # analogous to Input FPKMs.cor.test(x1, x2, method=c("spearman")) # will return no correlationplot(x1,x2) # same visuallycor.test(x1-y, x2-y, method=c("spearman")) # will return a big positive correlation of rho = 0.48, P < 2.2e^-16.plot(x1-y,x2-y) # same visuallyHere I used x1-y to represent the Log(X1/Y) because the data are log-normal log(X1) – log(Y) = log(X1/Y).So the result in Figure 2 (positive correlation between PABP occupancy and ME31B occupancy) could be spurious – coming from the common denominator input for both occupancies. The fact that this is not seen in Figure 2 suggests there are differences between the Embryo and S2 cell datasets, but it's very hard to interpret with this confounding denominator issue. Figure 6 shows decreased correlations over time, but these could be largely driven by lower correlations between input RNA-seq FPKMs (denominators) over time. Many results in the paper rely on this kind of analysis (occupancy vs. something else, where both are divided by mRNA levels (e.g. TE)). This potentially affects Figure 2, Figure 3, Figure 6, and 7. A biostatistician should be consulted to help sort out the best way to handle this. In addition, it would help to validate the RIP data (assuming it's clean see point 1) via northern blot or qRT-PCR for more genes than Act5C.

We thank the reviewer for this feedback and have addressed this point above.

4) Somewhat / potentially minor question and comment. The IP data in Figure 1 are interesting in that there is a second isoform of eIF4G that is abundant in S2 cells and present in the embryo lysate, but does not IP. What is this isoform and why is it not part of the eIF4G complex with PABP and eIF4E in embryos?

Like the reviewer, we also noticed the second isoform of eIF4G. Currently, we do not know the differences between the two isoforms, but we agree that this will be an interesting direction for future studies.

Reviewer #3:[…] However, importantly, these relationships are all at this stage simply correlative, and are not established as causal. The authors do explore the effects of two different factors on the trends they observe, and they nicely show the PNG kinase is critical for the loss of the ME31B, Cup, Tral proteins, but that SMG is not. But they do not connect directly the proteins that are the focus of the study, ME31B, Cup and Tral, to the phenomena that they describe through genetic approaches, as would be required to establish causality.

We thank the reviewer for raising this point. We agree that, with our current approaches, we cannot determine causality, and so we have rewritten the Discussion.

As a final but substantive point, I find the conclusions that the authors reach to be very much overstated for the type of data they have gathered. What is presented in this study, and in many previous studies of this nature, is a set of correlations that change during this time course of development. There are interesting events being explored and the data gathered are impressive. But the observation of such correlations does not permit strong statements of mechanism that depend on mutational analyses and careful ordering of events. As acknowledged by the authors in their Introduction, there are many changes happening during the MZT, including the activation of mRNA decay pathways that are silenced in the early embryo. Given that translation and mRNA decay are tightly coupled in so many systems, it seems at a minimum that these dependencies would be discussed as a possible (and indeed likely) explanation for what is observed in this system. And, since this possibility can't be eliminated by the analysis here, then it should be offered throughout as an alternative to the one preferred by the authors which is that the "mechanism" has changed from one of translational repression to one of mRNA decay during the MZT. Stated more clearly, if the decay machinery wasn't expressed early in the time course, then all that will be observed is translational repression, but if they are coupled later in the time course (as in much of biology), then the mRNA decay that happens will eliminate the signature of the translational repression. In the absence of mutations that separate these events, these alternative explanations must be offered as equally likely. In terms of simplicity, it seems easier to argue that translational repression is common throughout development, but that the increasing levels of decay machinery is a critical change that triggers the apparent change in "mechanism". The problem for the authors is that they see changes in protein levels (ME31B, Cup and Tral), but fundamentally, despite these changes, they are following the activities of mRNAs that still associate with these proteins.

We thank the reviewer for raising this point and apologize for not having dissected the different possibilities more carefully in our original submission. Although we still favor a model where Cup is important for blocking mRNA decapping in early embryos (and thus its loss enable clearance of maternal transcripts), we have no direct evidence to support this interpretation and so we have rewritten our discussion to make clear that this is just one view, among several possible models.

Similarly, the reviewer is correct that using “mechanism” overstates our conclusions, which rest on genetic evidence and temporal co-incidence. Accordingly, we have edited our manuscript to make clear that, currently, we cannot determine whether the lack of translational repression after 2–3 hr is because those transcripts are degraded (the most simple explanation) or because ME31B, mechanistically, does not repress translation. Like the reviewer, we believe that directly and rigorously dissecting the relationship between translational repression and mRNA decay during the MZT will be an important step in the future.